# Experimental Seismic Assessment of Nonstructural Elements: Testing Protocols and Novel Perspectives

Martino Zito [1,2], Roberto Nascimbene [2,3], Paolo Dubini [2], Danilo D'Angela [1,*] and Gennaro Magliulo [1,4]

1   Department of Structures for Engineering and Architecture, University of Naples Federico II, 80125 Naples, Italy
2   European Centre for Training and Research in Earthquake Engineering (EUCENTRE), 27100 Pavia, Italy
3   Department of Science, Technology and Society, University School for Advanced Studies IUSS Pavia, 27100 Pavia, Italy
4   Construction Technologies Institute (ITC), National Research Council (CNR), 80125 Naples, Italy
*   Correspondence: danilo.dangela@unina.it; Tel.: +39-329-9746213

**Abstract:** Nonstructural elements (NEs) are generally defined as elements typically housed within buildings/facilities that are not part of the structural system. Nonstructural elements are often classified as architectural elements, mechanical/electrical/hydraulic systems, and building contents. Nonstructural elements are often associated with critical seismic risk, due to their high vulnerability and exposure to seismic actions, especially for critical facilities such as hospitals and nuclear plant facilities. Accordingly, the combination of major exposure and vulnerability makes NEs extremely critical in terms of seismic risk even for low to moderate seismicity. The paper reviews and evaluates the main international testing approaches and protocols for the seismic assessment of NEs by means of experimental methods, which are referred to for seismic qualification. Existing test protocols are technically analyzed considering quasi-static, single-floor dynamic, and multi-floor dynamic procedures, supplying technical and operative guidance for their implementation, according to the latest advances in the field. The study proposes novel perspectives and a unified approach for the seismic assessment and qualification of NEs. The technical recommendations lay the groundwork for a more robust and standardized testing and qualification framework. In particular, the provided data might represent the first step for developing code and regulation criteria for the experimental seismic assessment and qualification of NEs.

**Keywords:** seismic assessment; nonstructural elements; seismic qualification; quasi-static testing; shake table testing; testing protocols



## 1. Introduction

Nonstructural elements (NEs) of a building consist of (building) elements/components and contents that are not included in the structural system of the facility. Nonstructural elements can be either single elements or distributed systems (e.g., networks). They are typically not intended to be load-bearing elements, and are often attached/connected to structural/building elements. Nonstructural elements can also be not connected to the structure/building and can be moved or relocated during their lifetime (e.g., furniture). The dynamic behavior of NEs is typically associated with high seismic risk, since NEs are often highly vulnerable and exposed to damage caused by seismic actions. The critical seismic response of NEs was highlighted by several recent earthquake events, e.g., the 2010 Darfield, 2011 Christchurch, and 2016 Kaikōura New Zealand earthquakes [1–3], the 2010 offshore Maule earthquake in Chile [4], the 2009 L'Aquila (Italy) earthquake [5], the 2012 Emilia (Italy) earthquake [6], the 2016 Central Italy earthquake [7], and the 2020 Petrinja (Croatia) earthquake [8]. In particular, these and other seismic events proved that damage to NEs can be significant even within buildings that exhibit minor or negligible structural damage, potentially resulting in major economic losses and casualties.

In most buildings, the biggest contributor to economic losses resulting from earthquakes is damage to NEs. In particular, they represent a large percentage of the total construction cost [9]. Figure 1 presents the cost distribution of four sample buildings including reinforced concrete (RC) residential buildings, hotels, office buildings, and hospitals, in the United States [9,10]. The cost of NEs is highest in hospitals, where approximately 92% of total construction costs are NEs; this number reduces to 87% for hotels, 82% for office buildings, and 60% for RC residential buildings. Furthermore, NEs are likely to exhibit moderate to severe damage even under relatively frequent earthquakes. Accordingly, the combination of major exposure and vulnerability makes NEs extremely critical in terms of seismic risk even over low to moderate seismicity areas.

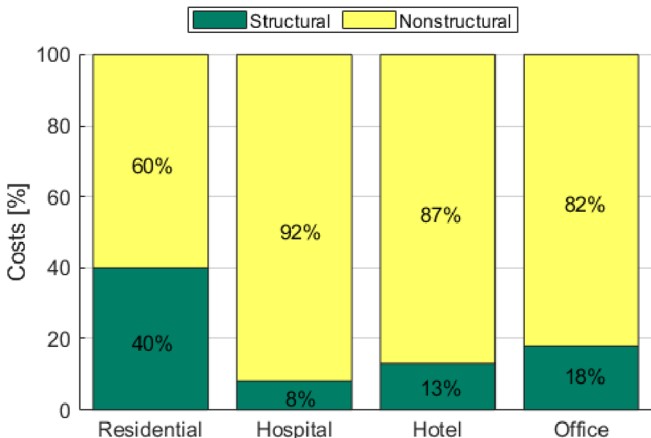

**Figure 1.** Cost breakdown of RC residential buildings, office buildings, hotels, and hospitals [9,10].

Over the last twenty years, the research community has been investigating the seismic assessment of NEs, and copious amounts of the literature have been developed accordingly. As a result, the current European and international codes and standards (e.g., [11–16]) refer to the Performance-Based Earthquake Engineering (PBEE) approach and methods for the design and assessment of NEs. Significant attention has been recently focused on the estimation of (building) floor accelerations for the assessment of the seismic demands on NEs in buildings [17–23]. Existing codes establish rules and criteria for the seismic design and verification of NEs considering the effects of the earthquake in terms of forces (i.e., accelerations) and displacements (i.e., deformations). The dynamic proprieties of NEs should be typically known, in order to apply criteria and methods established by the codes and standards for the determination of seismic demand measures, whereas the seismic capacities of NEs should be estimated in order to carry out the design and assessment safety verifications. The evaluation of the seismic capacities shall be carried out via one of the following methods: (a) analysis, (b) (experimental) testing, (c) experience data, (d) a combination of methods (a), (b), and (c) [13,24,25]. The seismic assessment of NEs, aiming at producing robust standard capacity estimations and safety evaluations, is often referred to as seismic qualification and, in some cases, seismic certification [13,24,26].

Several analytical/numerical models have been developed in the literature for the seismic evaluation of NEs. However, each study is typically associated with a specific NE and is not generally extendable to different NE properties and characteristics [27–33]. Experimental testing is the most common method used to assess NEs, as this is typically considered to be the most reliable and robust option. Many research activities are focused on the seismic assessment/qualification of NEs by experimental tests, such as on plasterboard partition walls [34–44]; masonry infill walls [45–49]; innovative partition walls [50–53]; bracing systems for suspended ceilings [54] and suspended ceiling systems [55–60]; curtain walls [61–64]; technical equipment and systems: motor control centers (MCC) [65], cooling machines [66], riser pipe systems [67], hospital piping assemblies [68], pressurized fire sprinkler piping systems [69], medical gas and fire-protection pipeline joints [70]; hospital

components [71–73]; museum artifacts and art objects [74–76], among many others. Seismic assessment and qualification through experience data might be theoretically more representative than other methods, since the NE response is estimated according to actual seismic events, but this is limited to specific characteristics of NEs, buildings, and seismic events. Regarding experimental test methods, testing protocols available in the literature generally vary depending on the type of NEs involved and the reference code. Finally, it is possible to develop hybrid methods to optimize the best characteristics of the single methods. Therefore, experimental testing is generally favored and considered to be more generally applicable and efficient, especially for critical and complex NEs.

The present paper discusses and evaluates the main international testing approaches and protocols, also considering available codes and guidelines for seismic qualification. Existing test protocols are subdivided into three categories: (1) quasi-static, (2) single-floor dynamic, and (3) multi-floor dynamic. In light of the above-mentioned evaluation, the study proposes a unified approach for the seismic assessment and qualification of NEs, to be possibly implemented in current regulations and technical guidelines. The technical and scientific novelty developed in the study fills a critical gap identified in the literature, and the technical recommendations and innovative perspectives potentially contribute towards a more reliable and robust seismic assessment of NEs.

Figure 2 depicts the outline of the paper and the flowchart of the methodology. In particular, Section 2 addresses NEs with regard to seismic damage, losses, and damage sensitivity. Section 3 provides the reference classification criteria in light of the NEs of interest. Section 4 describes, discusses, and evaluates the main testing method and protocols, with regard to the outcomes of the classification assessment and provides the key parameters and features of interest. Finally, Section 5 represents a technical commentary on the other sections, discussing the criticalities of existing protocols, potential improvement interventions, and technical recommendations and operative remarks. The details of Figure 2 are referred to within the relevant sections.

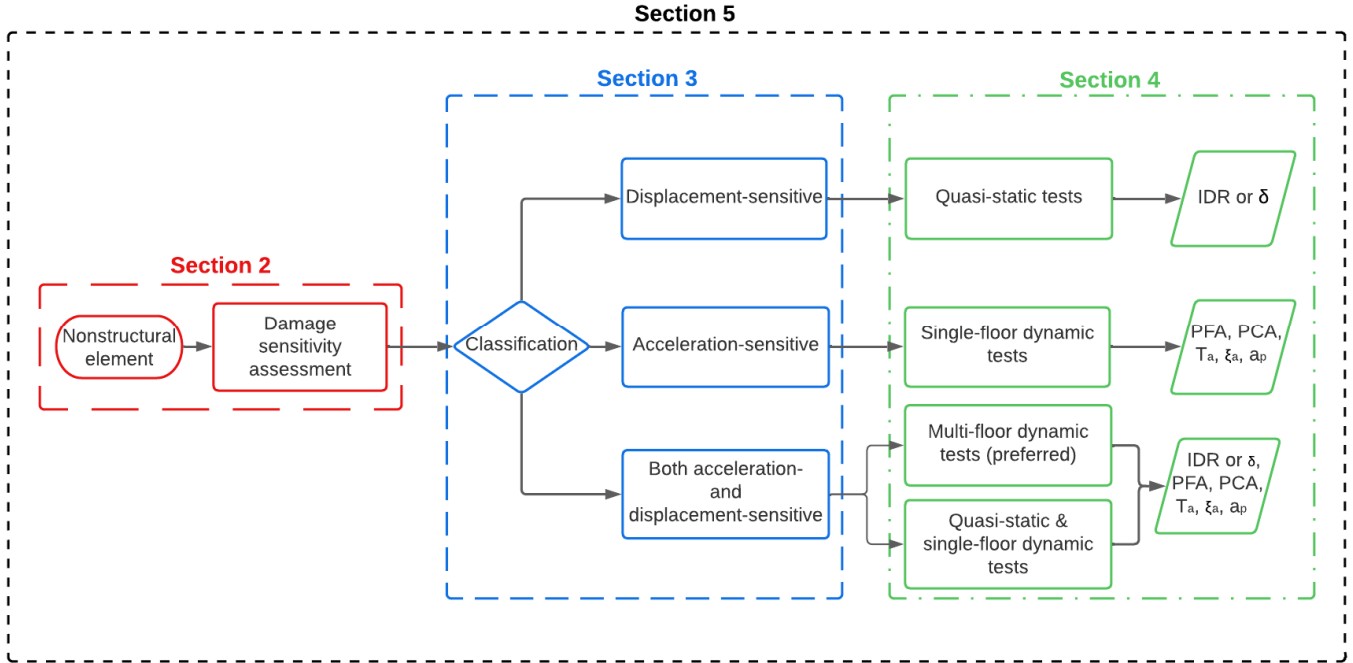

**Figure 2.** Outline of the paper and flowchart of the methodology.

## 2. Nonstructural Damage and Losses

Seismic events generally have four main effects on building NEs: (1) inertial or dynamic effects, (2) distortions imposed on NEs when the building structure deforms, (3) interaction at the interface between adjacent structures, and (4) interaction between

adjacent NEs [77]. Seismic damage to NEs is associated with several parameters and features, including the dynamic properties of the NEs, NE location within the building (e.g., along height), NE vicinity to structures or other NEs, NE vicinity to structure anchorage or connection systems and properties.

Nonstructural element damage may result in human losses and casualties, costly property damage to buildings and their contents, and functioning disruptions. In particular, the potential consequences of earthquake damage to NEs are typically divided into three types of risk referred to as the 3Ds: Deaths, Dollars, and Downtime [77]. The death type of risk is associated with the human losses caused by damage to or the response of NEs. Furthermore, life safety can also be affected in cases where damaged NEs obstruct safety pathways or exits [78]. Damage to life safety systems such as fire protection components or devices can also pose a safety concern in cases of fire after an earthquake or prior to fully restoring the system. Representative examples of critical and possibly hazardous NE damage include but are not limited to glass breaking, cabinets overturning, the collapse of ceilings and lighting system fixtures, the rupture of gas lines and other piping containing hazardous materials, damage to friable asbestos materials, the collapse of decorative molding parts, the failure of masonry infill walls, parapets, and chimneys (e.g., [9,79]).

The property losses (Dollars) may be the result of direct damage to a NE or the consequences produced by its damage [77]. The loss can be associated with private or public properties and might be particularly critical (even priceless) in cases of damage to museums/historical monuments/facilities/objects or archives/storage facilities. For example, the 2016 Central Italy earthquake caused major damage to the stuccoes and decorations of monumental churches and historical palaces [7], and the 2009 L'Aquila earthquake caused damage and even destruction of historically valuable sculptures located in the Spanish Fortress (L'Aquila) and damage to furniture and artifacts in a church [1].

Several earthquake events have highlighted that the functioning of buildings can be typically disrupted by damage to NEs even though structural damage is not exhibited (e.g., they were designed according to modern codes). For example, severe damage to masonry infills and partitions was observed after the 2009 L'Aquila earthquake, involving many buildings that were designed and constructed not much earlier than the earthquake; this caused significant losses in terms of Downtime [5]. Post-earthquake downtime is highly critical when the building/facility function is considered vital, especially in the aftermath of the event, e.g., hospital and healthcare facilities, fire and police stations, manufacturing facilities, and government offices. Furthermore, downtime has a huge impact in situations where seismic and other emergencies/crises (e.g., sanitary emergencies) are combined. For instance, in 2020, in Croatia, the combination of the COVID-19 pandemic and two destructive earthquakes had an extremely critical impact on the population in terms of physical, economical, and social/psychological burden [8].

## 3. Nonstructural Element (NE) Classifications

Nonstructural elements can be divided into three broad categories according to their service and function, namely: (1) architectural elements, such as infill and partition walls, curtain walls, ceiling systems, and architectural ornamentations; (2) mechanical, electrical, and plumbing elements for example pumps, chillers, fans, air-handling units, motor control centers, electrical cabinets, distribution panels, transformers, and piping; (3) furniture, fixtures and equipment, and contents such as shelving and bookcases, industrial storage racks, medical records, computer and desktop equipment, wall- and ceiling-mounted TVs and monitors, industrial chemicals and hazardous substances, historical and cultural objects [77].

Nonstructural elements are also generally classified in relation to one or more response parameters of the structure, namely engineering demand parameters (EDPs), and the related damage sensitivity of NEs [9]. Seismic actions potentially cause damage to (building) NEs according to four main modalities, as depicted in Figure 3: (a) inertial or shaking effects, causing component motion/oscillation, sliding, rocking or overturning; (b) building

deformations, damaging interconnected NEs; (c) building separations damaging NEs at the seismic building joints due to differential displacements or at the boundaries of the facility; (d) interaction between adjacent NEs having relative displacement/response [77]. According to the above-mentioned classification, NEs can be grouped into three classes: (a) force-sensitive or acceleration-sensitive NEs, where inertial forces or accelerations can be considered as main EDPs, (b) displacement-sensitive or interstory-drift-sensitive NEs, where relative displacements/deformations can be assumed as main EDPs, (c) combined force/displacement-sensitive NEs, where both inertial forces/acceleration and relative displacements/deformations can be considered as main EDPs. There might be cases in which NEs are particularly sensitive to alternative or additional EDPs, such as velocity, as can be observed with regard to rocking-dominated NEs [80]; therefore, these classifications represent the basic reference and might not be exhaustive over the wide range of NE scenarios, especially for components that exhibit complex seismic response, with regard to multiple response and damage mechanisms.

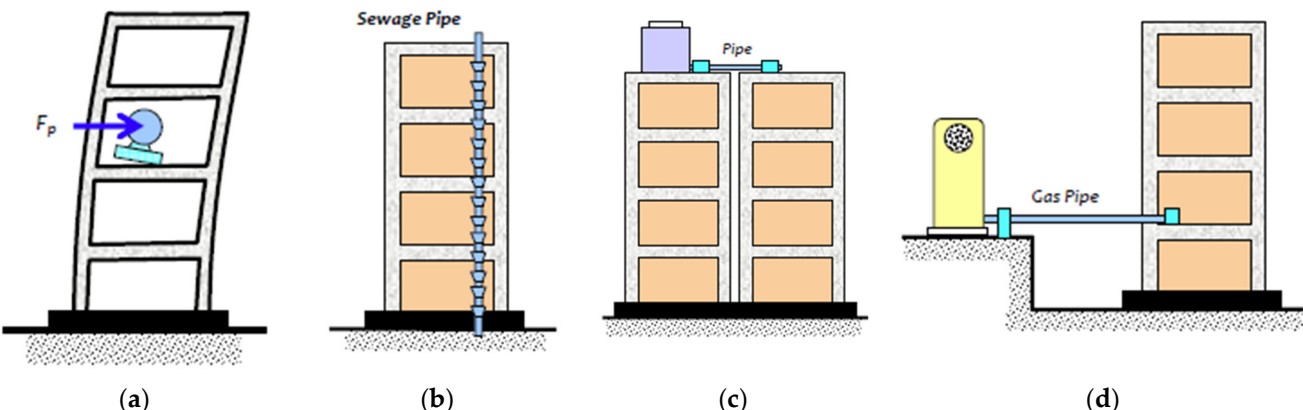

**Figure 3.** Effects of seismic motion on NEs: (**a**) inertial, (**b**) imposed deformations by building, (**c**) building separations, and (**d**) elements outside connected to the inside of the building [81].

Nonstructural elements can also be classified as a function of their dynamic parameters. In particular, ASCE 7-16 establishes in relation to the fundamental period of the NEs two types: flexible NEs, with a fundamental period larger than 0.06 s; and rigid NEs, with a fundamental period lower than or equal to 0.06 s. Finally, NEs can be defined according to the way they are built/assembled, e.g., the Italian code [12] defines two categories: built-on-site NEs and assembled/mounted-on-site NEs.

## 4. Testing Protocols

### 4.1. Outline

Seismic tests are typically classified as quasi-static or dynamic according to the testing procedure and protocol. Single-floor dynamic tests define the traditional and most common dynamic tests, which are often carried out by means of shake tables. More recently, multi-floor dynamic testing has been developed to replicate the actual seismic demands on NEs in a more accurate manner; in particular, this novel testing approach takes into account the response of the hosting buildings/facilities (in terms of both acceleration and deformation inputs) for the more representative and consistent seismic testing of NEs.

### 4.2. Quasi-Static Testing Protocols

Quasi-static tests are often performed to test displacement-sensitive NEs, or more generally, to test NEs that are sensitive to deformations. Nonstructural elements that can be tested according to this protocol include but are not limited to infill/cladding/partition panels (along their in-plane directions), piping and electrical network systems, fixed ladder systems, technical equipment fixed to multiple stories, and other systems that are connected to multiple building parts that might exhibit relative displacement under earthquake actions.

Quasi-static tests are typically carried out by implementing cyclic loading procedures, i.e., slow load/deformation cycles, which can be conducted in force or deformation control. Overall, it is preferred to adopt a deformation-controlled testing protocol, except when forces govern the response of the NE or the deformation parameter is difficult to control. Quasi-static protocols should not be considered for NEs whose behavior is significantly conditioned by dynamic effects or is velocity-sensitive, including strain–rate-sensitive NEs.

An example of the application of a quasi-static testing protocol is described in Pali et al. [37]. In this study, the authors designed a specific test setup for performing the in-plane cyclic tests on full-scale wall configurations according to the testing protocol described in Section 4.2.1. The test setup was built and assembled at the Test Laboratory of the Department of Structures for Engineering and Architecture at the University of Naples "Federico II", and a schematic of the testing arrangement/setup and instrumentation is depicted in Figure 4. The test setup was a bidimensional frame made of S350JR steel-grade hot-rolled steel profiles, which reproduced the behavior of a typical story of a building structure. It was made of a bottom beam, a top beam, and 2 hinged columns. In particular, the bottom beam was fixed to the laboratory floor, whereas the top beam was connected to the hydraulic loading actuator by means of a sliding hinge used for avoiding vertical load components. The loading actuator had a stroke of 500 mm and a load capacity of 500 kN. The bottom and top beams were connected to each other by means of 2 vertical columns having rectangular hollow sections. The connections between columns and beams were uniaxial hinges with axes of rotation perpendicular to the plane of the testing frame. The out-of-plane displacements of the testing frame were avoided by 2 steel portal frames, having HEA140 vertical profiles equipped with roller wheels that allowed the top beam of the testing frame to slide in its plane. Therefore, the testing frame had negligible lateral in-plane stiffness.

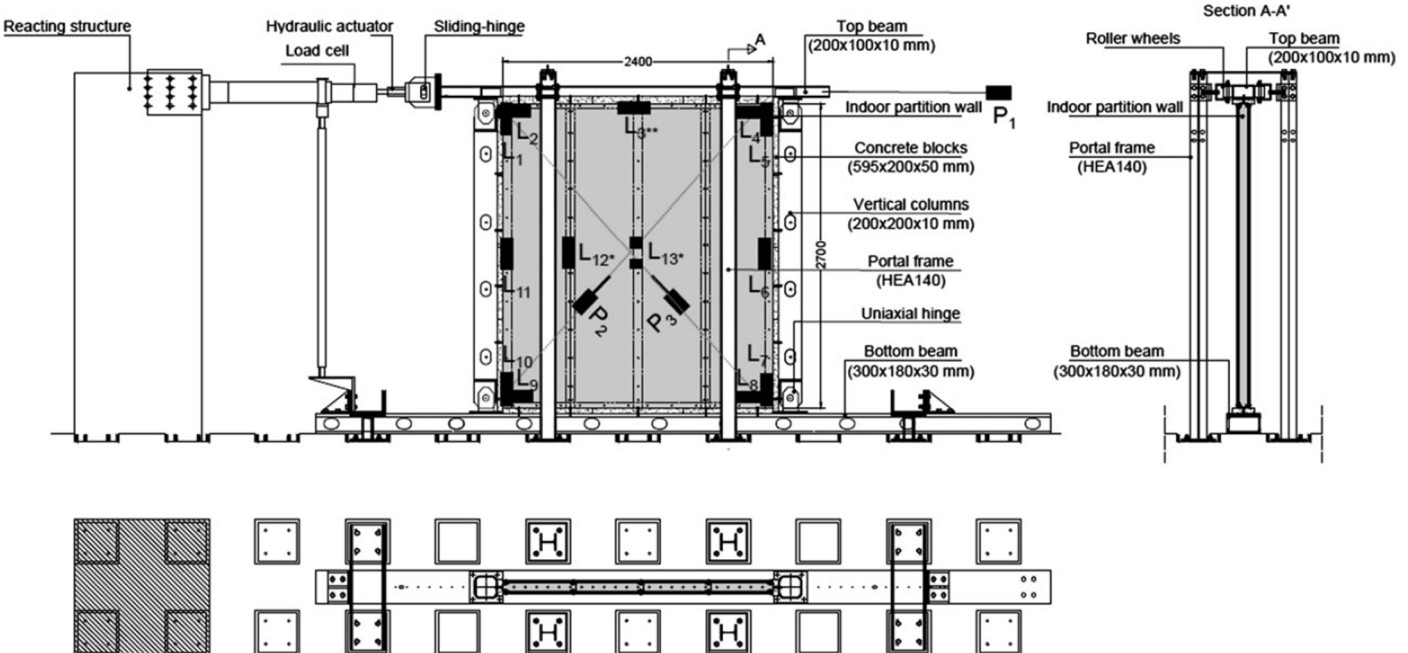

**Figure 4.** Typical test setup for quasi-static cyclic testing of the drywall partitions [37].

The indoor partition wall was infilled in the testing frame in such a way that it was surrounded along its perimeter by the top beam, bottom beam, and columns. Moreover, the setup was designed to allow the interposing of C25/30 strength-class 50 mm thick concrete blocks between specimens and testing frame, in such a way as to simulate the connections with a reinforced concrete building structure. In particular, concrete blocks were placed on the faces of beams (horizontal connections) and columns (vertical connections) of the testing frame facing the indoor partition wall.

### 4.2.1. FEMA 461

The FEMA 461 report [82] provides a protocol for quasi-static cyclic testing of structural members and NEs. The testing protocol provides technical guidance for the achievement of the following sequential objectives: (a) identification of relevant damage states (DSs); (b) identification of EDPs that are well correlated with DSs identified in (a); and (c) the testing of NEs according to a well-defined testing plan and loading protocol to establish quantitative correlations between DS achievement and EDP thresholds. The protocol provides information regarding the following technical aspects: (1) procurement, fabrication, and inspection of testing specimens; (2) extrapolation and interpolation of similar components; (3) laboratory standards, including accreditation criteria, actuators, instruments, data acquisition systems, and safety procedures; (4) test plan and procedures; (5) testing directions and loading control parameters; (6) loading histories, including unidirectional testing, bidirectional testing, and force-controlled loading; and (7) reporting. The FEMA 461 protocol could be used to assess fragility data and to estimate relevant response properties and hysteretic information. In the following, deformation- and force-controlled testing protocols are described.

#### Deformation-Controlled Testing Protocol

Deformation-controlled tests are typically preferred to force-controlled ones since deformation is often better correlated than force with seismic response of displacement-sensitive elements; in particular, differently from force-controlled procedures, deformation-controlled testing typically allows the assessment of the inelastic response, with particular regard to plastic and degrading behavior. The deformation control EDP may be a displacement or other suitable deformation quantity, e.g., a rotation. This parameter should be correlated with a building deformation parameter, such as interstory drift ratio (IDR), that can be estimated, in terms of seismic demand scenarios, by structural analysis. The testing deformation increment should be selected in a manner that ensures reliability and robustness in the experimental test. The deformation increment should be sufficiently small that: (a) dynamic effects are negligible; (b) the value of the deformation parameter (associated with DSs of interest) can be clearly identified; (c) thermal effects related to work-hardening are negligible; and (d) power requirements are reasonable. Moreover, the deformation increment should be sufficiently large that: (a) the duration of the test is not excessive; (b) material creep is not a significant effect; and (c) the number of cycles experienced by the component at the onset of significant DSs is comparable with realistic ones under strong earthquake response. Low-cycle fatigue response should not be accounted for since this is not typically observed in real building elements.

The parameters required to define the loading history in terms of deformation-controlled testing protocol are the smallest targeted deformation amplitude of the loading history ($\Delta_0$), maximum target deformation amplitude of the loading history ($\Delta_m$), number of steps (or increments) in the loading history (n), generally 10 or larger, and amplitude of the cycles (ai). $\Delta_0$ must be safely smaller than the amplitude at which the lowest significant DS is first observed; a recommended value (in terms of IDR) is 0.0015. $\Delta_m$ is an estimated value of the imposed deformation at which the most severe damage level is expected to initiate; a recommended value for this amplitude (in terms of IDR) is 0.03. The loading history consists of repeated cycles of step-wise increasing deformation amplitudes. Two

cycles at each amplitude shall be completed. The amplitude $a_{i+1}$ of the step $i + 1$ is given by Equation (1):

$$a_{i+1} = c\, a_i, \tag{1}$$

where $a_i$ is the amplitude of the $i^{th}$ and $c$ is a parameter suggested to be assumed equal to 1.4.

In case the failure (or relevant final damage condition) is not achieved corresponding to $\Delta_m$, constant increments 0.3 $\Delta_m$ can be applied to extend the loading history by increasing further the testing amplitude. Figure 5a shows the loading protocol corresponding to $a_n = \Delta_m$, $a_1 = 0.0048\,\Delta_m$, and n = 10. In case bidirectional testing has to be considered, an elliptical loading pattern should be applied, as it is depicted in Figure 5b.

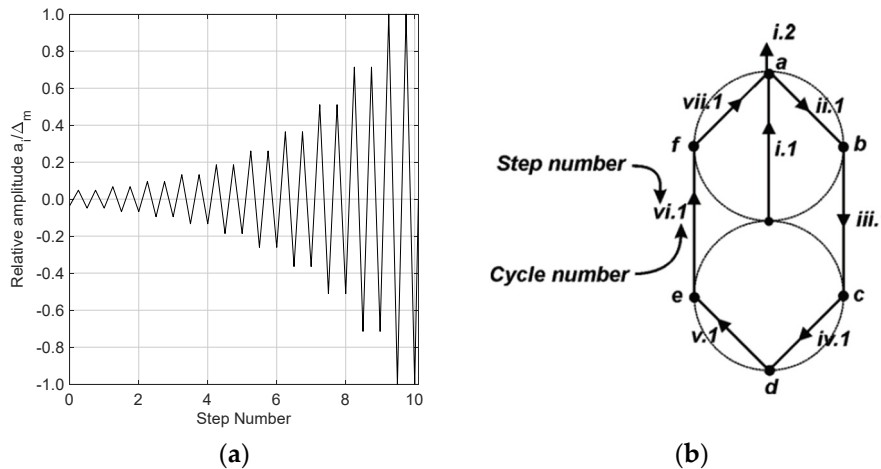

(a)  (b)

**Figure 5.** FEMA 461 displacement-controlled quasi-static protocol: (**a**) loading history considering $\{a_n, a_1, n\}$ equal to $\{\Delta_m, 0.0048\,\Delta_m, 10\}$ and (**b**) displacement orbit for bidirectional loading test.

Force-Controlled Testing Protocol

Force-controlled testing should be performed if the performance of NE can be correlated with force measures, or if a suitable deformation parameter cannot be found/assessed. The force-controlled EDP shall be a measurable and controllable parameter (typically, a force) that is compatible with a (force-based) measure that can be assessed via structural analysis. The force increment should be sufficiently small that: (a) dynamic effects are negligible; (b) the applied force initiating the various damage states of interest must be identifiable; (c) thermal effects due to work hardening are not significant; and (d) power requirements are not unreasonable. Finally, the force increment should be sufficiently large that: (a) the duration of the test is not excessive and (b) material creep is not a significant effect (unless creep is considered to be part of the damage states of interest).

The reference value on which to base the amplitudes of individual cycles is the maximum force to which NE (or part) may be subjected in a severe earthquake. Since the force demands strongly depend on characteristics/features of NE, building to NE interaction, building, and site, it is impossible to develop a generally applicable force-based loading protocol. Therefore, the following guidelines should be employed to develop case-specific protocols.

Forces are consequences of deformations, and the deformations, in relative magnitude, can be described by Equation (1) and Figure 5a. If the monotonic force–deformation response of the force-sensitive NE is known (e.g., Figure 6a), then the displacement-based loading history (Figure 5a) and the NE response can be combined to develop a force-based loading history to be applied to NE (Figure 6b).

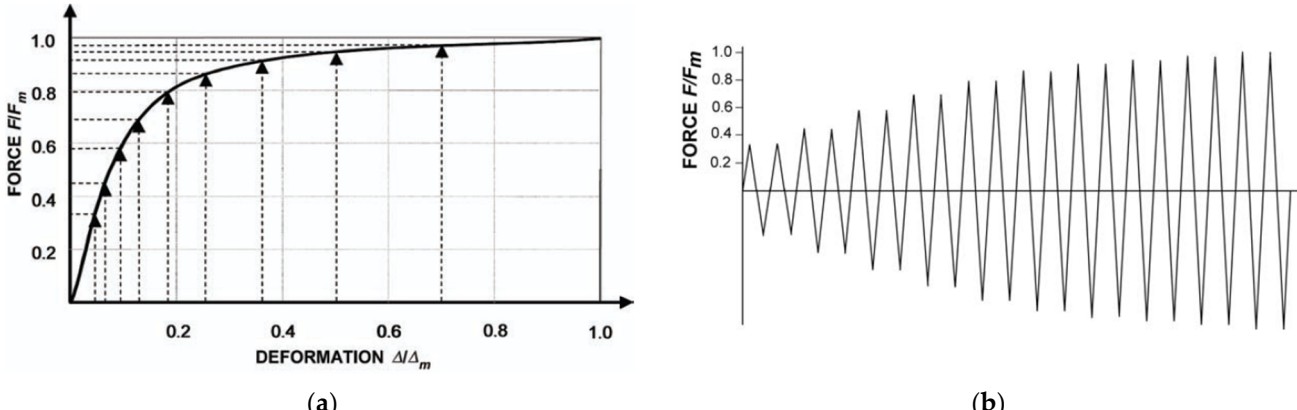

**Figure 6.** FEMA 461 force-controlled quasi-static protocol: (**a**) force-displacement NE response and (**b**) derived loading history.

### 4.2.2. CUREE-Caltech

The CUREE–Caltech testing protocol [83] was developed by the Consortium of Universities for Research in Earthquake Engineering (CUREE) and California Institute of Technology (Caltech) in the framework of the extensive CUREE–Caltech Woodframe Project. This document provides recommendations for a protocol for quasi-static experimentation on components of wood-frame structures. Testing protocols typically define the construction and instrumentation of test specimens, the planning and execution of experiments, the loading history to be applied to a test specimen, and the documentation of experimental results. The CUREE–Calthech protocol gives significant importance to the development of loading histories for force- and deformation-controlled tests. The provided data are based on the findings of inelastic time history analysis of hysteretic systems under a wide range of ground motions (ordinary and near-field records). The protocol loading inputs are derived by processing the above-mentioned results through damage accumulation concepts and criteria.

The primary aim of this loading protocol is to evaluate the capacity level seismic performance of NEs subjected to ordinary earthquake records with a probability of exceedance equal to 10% within 50 years. Loading histories associated with smaller events prior to the capacity level event are also accounted for by the deformation history. Similarly to FEMA 461 protocol, deformation-controlled procedures should be preferred to force-controlled ones, and the latter should be used only for NEs whose behavior is controlled by forces rather than deformations or if a suitable deformation EDP has not been found, which, in general, corresponds to NEs governed by brittle failure modes; both deformation- and force-controlled protocols are defined.

Figure 7 shows the loading protocol associated with a representative cyclic load test. The protocol input is defined by variations in deformation amplitudes, considering the reference deformation $\Delta$ as an absolute measure of deformation amplitude. The history consists of initiation, primary, and trailing cycles. Initiation cycles are intended to check loading equipment, measurement devices, and the force–deformation response at small amplitudes. A primary cycle is a cycle that is larger than all the preceding cycles and is followed by smaller cycles, which are called trailing cycles. All trailing cycles have an amplitude that is equal to 75% of the amplitude of the preceding primary cycle. All cycles are symmetric in terms of positive and negative amplitudes. Deformation control should be considered throughout the experiment.

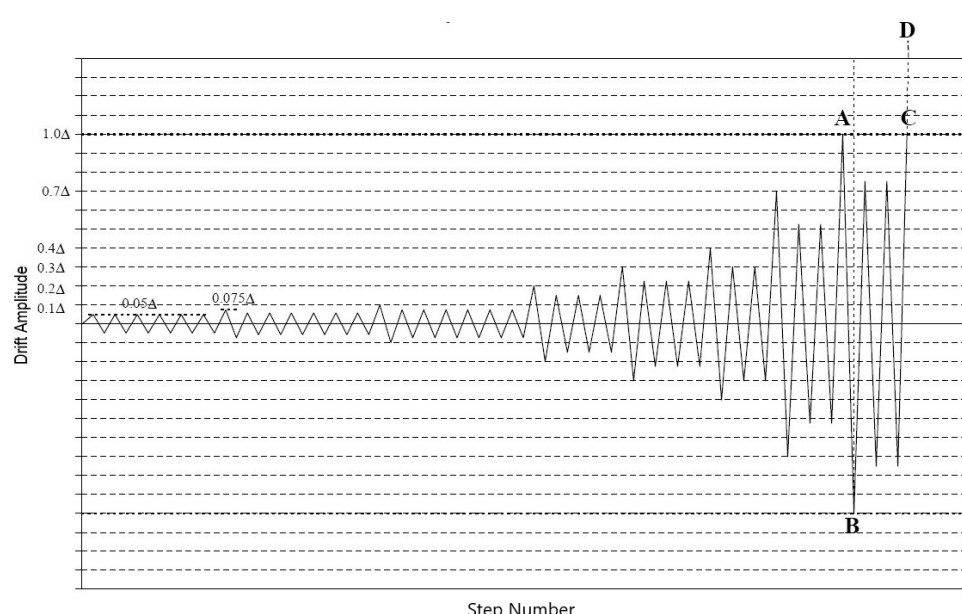

**Figure 7.** Loading history for basic deformation controlled quasi-static cyclic test [83].

The reference deformation is the maximum deformation that NE is expected to sustain according to a target criterion and assuming that the proposed basic loading history has been applied to the test specimen. This is an expected measure of the specimen deformation capacity, which should be estimated prior to performing the tests. This capacity could be assessed according to past data or previous experience, monotonic tests, or a consensus value that may prove to be useful for comparing tests of different details or configurations. The reference threshold $\Delta$ may depend on the specific element to test or may be fixed for a specific testing program according to the following steps. (1) Performing monotonic tests, which provide data on the (monotonic) deformation capacity, $\Delta m$; this capacity corresponds to the deformation at which the load is reduced by 20% from the maximum applied load, as depicted in Figure 8 [83]. (2) Using a specific fraction of $\Delta_m$, i.e., $\gamma\Delta_m$, as the reference deformation for the basic cyclic load test. The factor $\gamma$ accounts for the different deformation capacity between the monotonic and the cyclic testing procedures; $\gamma$ is suggested to be assumed equal to 0.6.

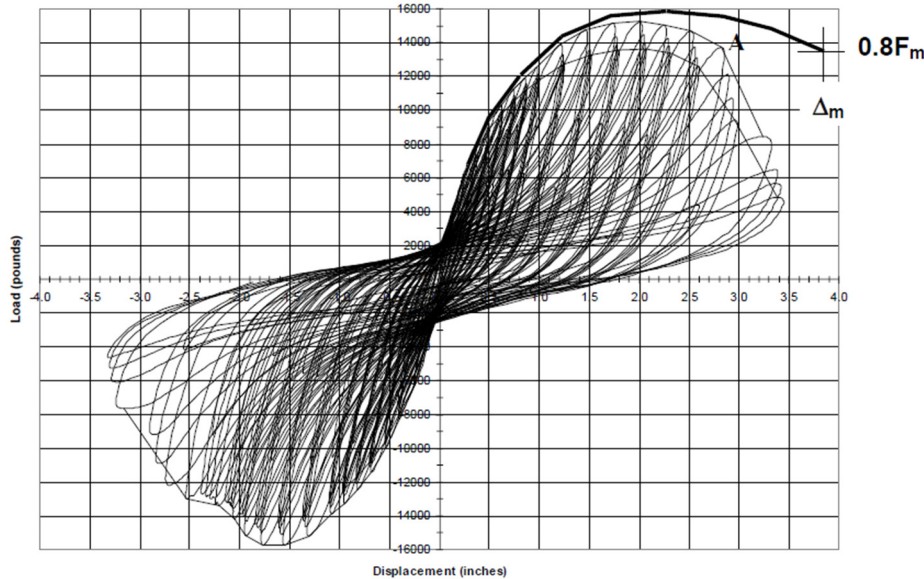

**Figure 8.** Definition of $\Delta m$ and its relation to a cyclic test [83].

The CUREE protocol proposes three other versions of the loading history protocol: (1) an abbreviated version of the basic protocol input, based on a smaller number of cycles; (2) a simplified version of the basic protocol input, having the same amplitude as the trailing cycles of preceding primary cycle; and (3) a protocol input associated with near-fault records (i.e., seismic hazard with a 2% probability of exceedance within 50 years).

### 4.2.3. AAMA 501.4 and 501.6

The American Architectural Manufacturers Association (AAMA) has developed two test methods: AAMA 501.4 and 501.6 [84]. The AAMA 501.4-01 test provides a methodology for evaluating the performance of curtain and storefront walls under specified horizontal displacements along their in-plane direction. The method does not account for dynamic, torsional or vertical response. This method is complementary to AAMA 501.6, which considers ultimate limit state for architectural glass included in the wall or partition system, accounting for fallout from window wall, curtain and storefront walls. Differently, AAMA 501.4 aims to address the seismic serviceability limit state for wall systems and relevant changes or variations in terms of functioning/operation (e.g., rates of air/water leakage), as a result of statically applied, in-plane (horizontal) racking displacements.

The AAMA 501.4 test method establishes the procedures to assess the performance of curtain and storefront systems under laboratory conditions, when subjected to horizontal displacement intended to represent the effects of an earthquake or a significant wind event. The design displacement shall be determined according to the predicted interstory deformation of the subject building. For multistory setups, the displacement between levels may vary according to different story heights. Unless otherwise specified, the design displacement shall be 0.010 times the largest adjacent story heights. The displacement should not be measured at the specimen but corresponding to the movable floor element. Prior to conducting the displacement tests, the test specimen shall be, at a minimum, assessed for serviceability by conducting air leakage and water penetration tests. The air leakage and water penetration tests shall be conducted at the differential pressures specified for the application/functioning conditions. Moreover, the air leakage and water penetration tests are repeated after the displacement tests to evaluate the change in functionality of the wall system. In this case, the static displacement is evaluated considering a value equal to 1.5 times the design displacement used in the previous step.

The AAMA 501.6 test method is applicable to any type of glass panel installed within wall system framing members, including the associated glazing elements (setting blocks, gaskets, fasteners, etc.). In the design of a building wall system to resist earthquakes, in case glass-to-frame contacts cannot be avoided, or if the glass elements are not highly resistant to earthquake-induced glass fallout, ASCE calls for the determination of $\Delta_{fallout}$, i.e., the relative seismic displacement (drift) causing glass fallout from the curtain wall, storefront or partition; $\Delta_{fallout}$ should be estimated through dynamic tests. In particular, this testing method aims at defining a dynamic racking crescendo test for determining $\Delta_{fallout}$. According to the ASCE 7-16, dynamic tests are not required when adequate clearance exists between glass edges and wall frame glazing pockets to prevent contact/interaction under seismic design displacements in the main structural system of the building.

The AAMA 501.6 test method involves mounting individual fully glazed wall panel specimens on a dynamic racking test apparatus; the test loads the specimen through sinusoidal racking motions according to gradual progressive increasing racking amplitudes (Figure 9). The dynamic racking frequencies are equal to 0.8 Hz at lower racking amplitudes (i.e., $\leq \pm 75$ mm) and equal to 0.4 Hz at higher racking amplitudes (i.e., $> \pm 75$ mm). The racking amplitude associated with earliest glass fallout is assumed to define the fallout condition for that test specimen. The lowest value of racking displacement causing glass fallout for the three specimens tested by AAMA 501.6 is the reference value of fallout for that particular wall system glazing configuration. This value of fallout is referred to by ASCE 7-16 for architectural glass.

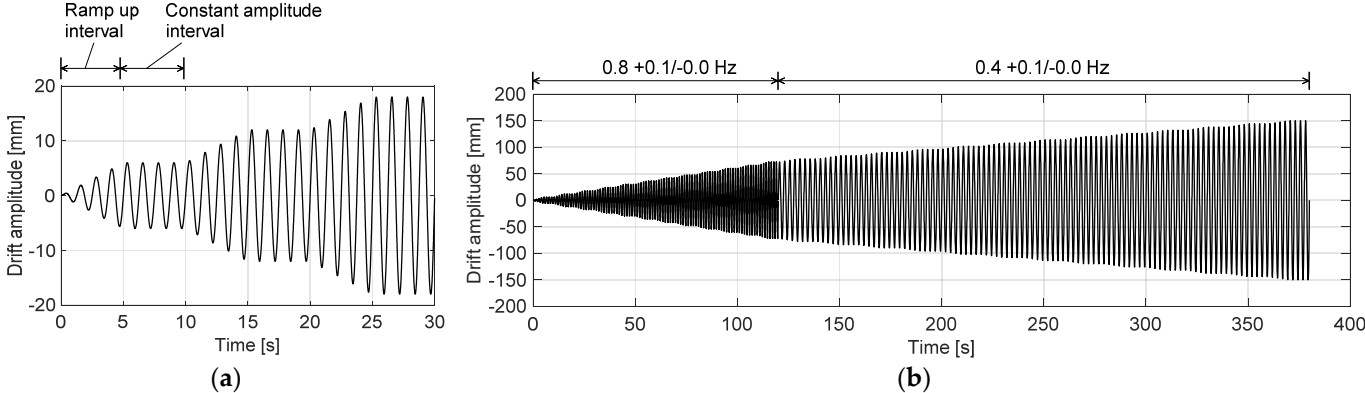

**Figure 9.** Drift time history for dynamic crescendo test: (**a**) first 30 s and (**b**) full time history (AAMA 501.6 2002) [84].

### 4.3. Single-Floor Dynamic Testing Protocols

Single-floor dynamic tests are frequently performed to test acceleration-sensitive NEs. Examples of NEs that may be tested in accordance with this protocol include cabinets, storage racks, bookcases and shelves, appliances (refrigerators, washing machines, diesel generators), hoardings anchored on rooftops, antennas (communication towers on rooftops), horizontal projections (sunshades, canopies, and marquees), storage vessels, mechanical equipment (boilers and furnaces, HVAC equipment), hospital cabinets, and museum artifacts.

Single-floor dynamic tests are generally performed through a shake table or earthquake simulator, which is usually characterized by one or more degrees of freedom (translational and rotational). Tests are conducted by assigning input signals to the earthquake simulator, which depending on different protocols may be generated differently (artificial or real earthquakes). Earthquake simulators often have performance limitations in terms of acceleration, velocity, displacement, and frequency range, so the assigned input signals must be adjusted/modified to be compatible with these limitations. However, the input signals must meet the spectral compatibility conditions required by the testing protocols.

An example of the application of a single-floor dynamic testing protocol is described in Truong et al. [85]. In this paper, the authors carried out shaking table tests for seismic performance evaluation of a base-isolated Uninterruptible Power Supply (UPS). The tests were performed according to the protocol described later, in Section 4.3.2.

Figure 10 presents the shaking table test setup for the UPS test specimen. The shake table used for the triaxial test has the following primary characteristics: 4.0 × 4.0 m plan dimensions, six degrees of freedom, a maximum payload of 300 kN, peak accelerations of 1.5, 1.5, and 1.0 g in the X, Y, and Z directions, respectively, for the maximum payload, and a maximum overturning moment of 1200 kNm. The shake table reproduces earthquake input ground motions through a system of eight hydraulic actuators. The UPS test specimen was anchored to an RC slab using pins or through isolator systems with anchor bolts. The concrete slab was connected to the shaking table using anchor bolts.

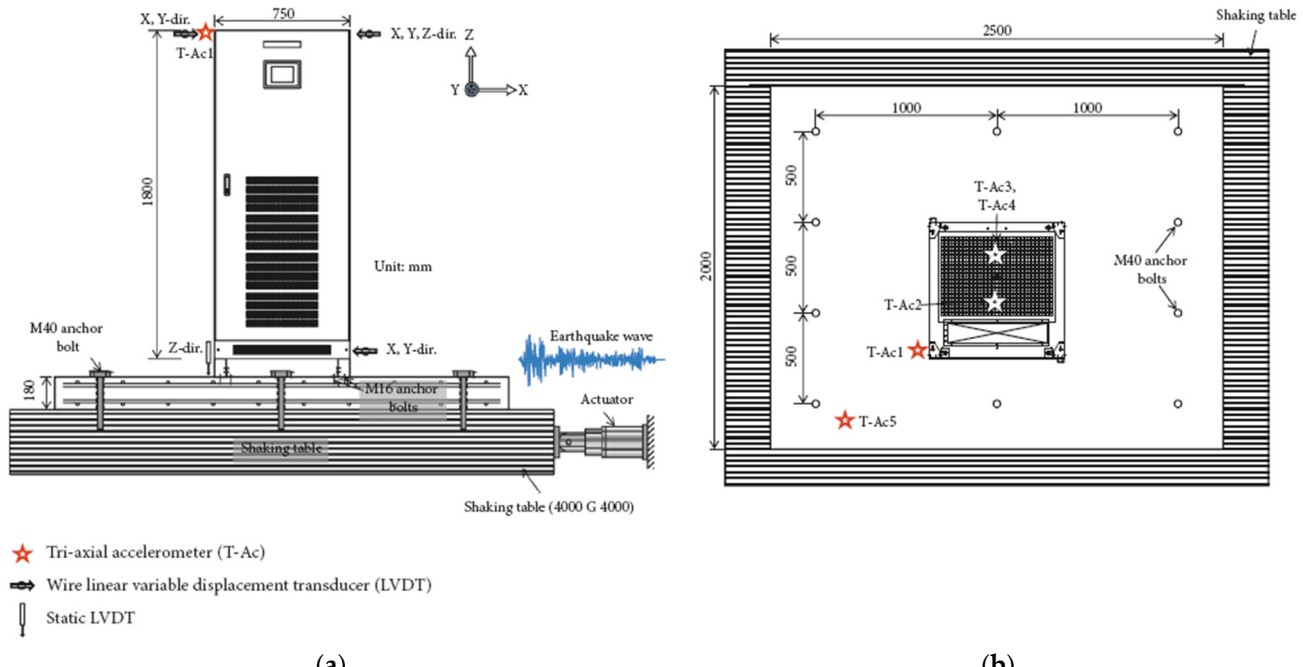

**Figure 10.** Typical shaking table test setup: (**a**) front view and (**b**) plan view [85].

### 4.3.1. FEMA 461

The FEMA 461 test method [82] defines a protocol for shake table testing of NEs to determine fragilities according to the PBEE approach. This protocol is intended to assess the seismic performance of NEs whose behavior is affected by the dynamic response of the component itself, or whose behavior is velocity-sensitive, or sensitive to strain–rate effects. The protocol includes the procedures and types of testing, test plan, input motion, test equipment and instrumentations, and information on the test report. The first step of the testing procedure includes (1) preliminary inspection and functional verification of the test specimen and (2) definition of functional performance and damage states. In particular, before testing, the test specimen should be examined to verify its functional performance, and appropriate DSs should be defined. Finally, a preliminary estimate of the excitation frequency and intensity should be carried out, considering the peak spectral accelerations corresponding to the component natural frequency, associated with relevant DSs.

The test plan of the FEMA 461 protocol includes three types of tests: (1) system identification tests; (2) seismic performance evaluation tests; and (3) failure tests. System identification tests should be conducted to identify the dynamic characteristics of the test specimen, considering the undamaged conditions (prior to the seismic performance evaluation test) and the evolution along the damage evolution (along the seismic performance evaluation test program). The dynamic properties to estimate include natural frequencies, equivalent fundamental modal viscous damping ratios, and mode shapes. In particular, FEMA 461 protocol establishes that identification tests should be conducted along each principal specimen direction prior to and following each performance/failure test. The dynamic identification tests can be carried out by selecting a test type among four types of tests: white noise tests, single-axis acceleration-controlled sinusoidal sweep tests, resonance tests, and static pull-back tests.

Seismic performance evaluation tests aim at assessing the performance of the specimen with regard to minor to moderate damage conditions, considering an artificial acceleration input and according to an incremental intensity procedure. Failure tests consist of seismic evaluation tests carried out considering higher intensities, in order to assess severe damage and (incipient) failure of the specimen or to assess DSs associated with potential safety

risk. Seismic performance evaluation tests and failure test descriptions and related results should be documented for each intensity level to provide data for fragility assessment.

The peak spectral acceleration corresponding to the natural frequency of the specimen should be considered as a primary input motion parameter associated with seismic intensity. The FEMA 461 protocol establishes that at least three different shaking intensities should be used for the performance evaluation test, and in all cases, a 25% increase in intensity should be the minimum step size between intensity levels. Moreover, the input motions of the performance evaluation and failure tests should be applied along the principal axes of the test specimen by performing triaxial tests or biaxial tests (along a horizontal and vertical direction) considering both horizontal directions (double biaxial tests); if the effect of vertical motion on the seismic response of the test specimen is negligible, biaxial horizontal tests can be performed. The procedure for generating compliant seismic inputs and provided inputs is based on work done by the U.S. Army Construction Engineering Research Laboratory [86]. In particular, the recommended seismic inputs consist in narrow-band random sweep acceleration signals, having scaled amplitudes depending on the sweep frequency, producing motions that have relatively smooth response spectra amplitudes. For records generated for this protocol, the bandwidth is one-third-octave, and the center frequency of the records sweeps from 0.5 Hz up to 32 Hz, at a rate of 6 octaves per minute or a total signal duration of 60 s. The vertical motion is scaled to have a response spectrum that is approximately 80% of horizontal ones. The response spectra should be defined considering 5% damping. The signal shall be scaled in order to have (a) acceleration response spectra amplitude equal to about 1 g within 2–32 Hz and (b) the displacement response spectra would be approximately uniform below 2 Hz. Figure 11 shows the shake table motions as they are provided by FEMA 461: (a) horizontal (longitudinal and transversal) and vertical acceleration time histories and (b) related response spectra (5% damped).

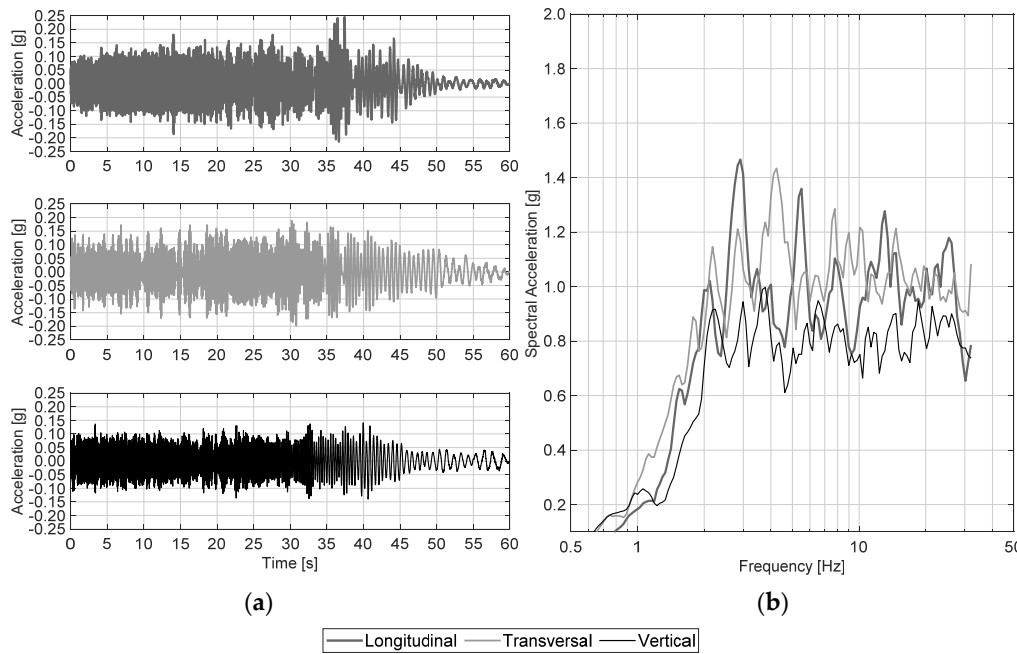

**Figure 11.** Recommended shake table motions according to FEMA 461: (**a**) horizontal (longitudinal and transversal) and vertical input motions and (**b**) acceleration response spectra, 5% damped.

### 4.3.2. ICC-ES AC156

The aim of AC156 testing protocol [26] is to establish criteria for seismic qualification/certification of NEs using shake table tests. In particular, these acceptance criteria are applicable for architectural, mechanical, electrical, and other nonstructural systems, components, and elements anchored to structures. This protocol is applicable for shake

table testing of NEs that have fundamental frequencies greater than or equal to 1.3 Hz and is not aimed to assess the effects of relative displacements of NEs. AC156 also reports the formal requirements for issuing the seismic certification.

The testing protocol should be generally performed along the three principal directions. When it is not possible to conduct triaxial testing due to facility limitations, biaxial or uniaxial tests may be conducted by the following guidelines. (1) Biaxial tests shall be performed in two phases. One of two horizontal shakings and the vertical shaking of the specimen for a fixed specimen configuration shall be considered in the first phase of the test; in the second phase, the same specimen must be rotated 90 degrees about the vertical axis. (2) Uniaxial tests shall be performed along the three distinct directions, with the test specimen rotated after each phase, such that all three principal axes of the specimen are considered.

AC156 defines two types of tests: resonant frequency search tests and seismic simulation tests. The former are for determining the resonant frequencies and damping ratios along each orthogonal axis of the test specimen, whereas the latter are for seismic performance evaluation and certification purposes. The input signal of the resonant frequency search test shall be a low-level amplitude single-axis sinusoidal sweep from 1.3 to 33.3 Hz. In particular, the peak input should be $0.1 \pm 0.05$ g, but a lower input level can be used to avoid specimen damage. Finally, the sweep rate shall be two octaves per minute, or less, in order to guarantee a sufficient time for maximum response at the resonant frequencies. The input signal for seismic simulation tests shall be determined for replicating the combined effects of the horizontal and vertical earthquakes. Required response spectra (RRS) associated with both horizontal and vertical directions are shown in Figure 12; RRS shall be developed according to the total design horizontal force, $F_p$, provided by ASCE 7-16, shown in Equation (2):

$$\frac{F_p}{W_p} = \frac{0.4\,S_{DS}\,a_p}{\frac{R_p}{I_p}}\left(1 + 2\frac{z}{h}\right),\tag{2}$$

and considering 5% damping.

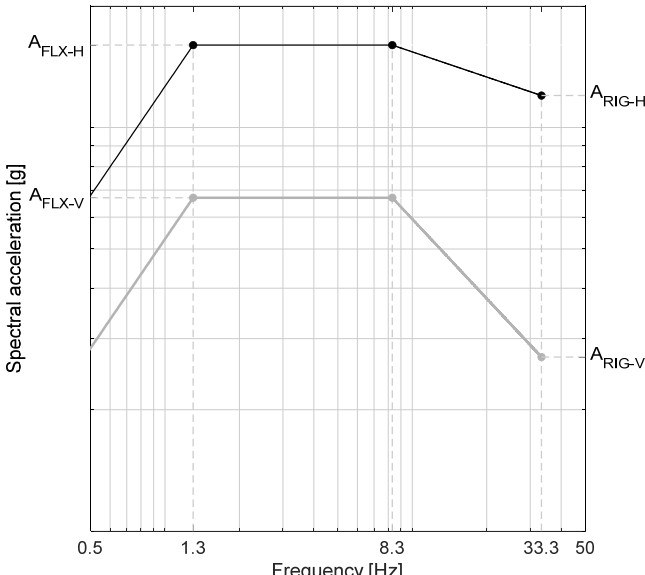

**Figure 12.** Required response spectra in the horizontal (black) and vertical (grey) directions according to AC156, 5% damped.

In Equation (2), $F_p$ is total design horizontal force applied in the center of mass or distributed according to the mass distribution of the NEs; $W_p$ is the weight of the NE; $R_p/I_p$ is the ratio of the component response modification factor $R_p$ to the component importance

factor $I_p$. $R_p/I_p$ represents a design reduction factor accounting for inelastic response and allowable inelastic energy absorption capacity of the system; $a_p$ is the component amplification factor that varies from 1.00 to 2.50; $z$ is the height in the structure of the point of attachment of the element concerning the base; $h$ is the average roof height of the structure for the base. The height factor ratio $z/h$ accounts for above grade level component installations within the primary supporting structure and ranges from zero at grade level to one at roof level, essentially acting as a force increase factor to recognize building amplification as you move up within the primary structure; $S_{DS}$ is the design spectral response acceleration parameter at short periods, while the product 0.4 $S_{DS}$ represents the point at period T = 0 s of the design response spectrum. $S_{DS}$ varies depending on geographic location and site soil conditions ASCE 7-16 [13].

The test specimen might exhibit an inelastic behavior under seismic tests. Therefore, the ratio $R_p/I_p$ shall be generally set equal to 1, corresponding to an unreduced response. $a_p$ represents a force increase factor by implementing amplification of response associated with the deformability of the NEs; this parameter shall be taken from the formal definition of flexible and rigid elements. By definition, NE is considered flexible (maximum amplification $a_p$ = 2.5) for fundamental frequencies less than 16.7 Hz, corresponding to the amplified region of the RRS. For fundamental frequencies greater than 16.7 Hz, NE is considered rigid (minimum $a_p$ = 1.0), corresponding to the zero peak acceleration (ZPA) range. This results in two normalizing acceleration factors that, when combined, define the horizontal RRS, as reported in Equations (3) and (4):

$$A_{RIG-H} = 0.4\, S_{DS}\left(1 + 2\frac{z}{h}\right), \tag{3}$$

$$A_{FLX-H} = S_{DS}\left(1 + 2\frac{z}{h}\right) \leq 1.6 S_{DS}, \tag{4}$$

where, $A_{FLX-H}$ should not exceed 1.6 $S_{DS}$.

The RRS for the vertical direction is associated with two-thirds of the ground-level base horizontal acceleration. Moreover, $z/h$ could be assumed equal to zero for all attachment heights, as is reported in Equation (5):

$$A_{RIG-V} = \frac{2}{3}A_{RIG-H} = 0.27 \cdot S_{DS};\ A_{FLX-V} = \frac{2}{3}A_{FLX-H} = 0.67 \cdot S_{DS}. \tag{5}$$

The input signals for seismic simulation tests shall be nonstationary broadband random excitations having an energy content ranging from 1.3 to 33.3 Hz, and a bandwidth resolution equal to one-third or one sixth-octave, depending on whether the synthesizer is analog or digital, respectively. The total duration of the input motion shall be 30 s, with the nonstationary character being synthesized by an input signal build-hold-decay envelope. The build time includes the time necessary for acceleration ramp-up, the hold time represents the earthquake strong-motion time duration, and the decay time includes the deceleration ring down time. At least 20 s of strong motion should be contained by the signal.

AC156 establishes the criteria and rules for the analysis of the test response spectrum (TRS) with regard to RRS, i.e., spectrum compatibility criteria. The TRS shall be computed using either justifiable analytical techniques or response spectrum analysis equipment using the control accelerometers located at the specimen base. The TRS shall be calculated using a damping value equal to 5% of critical damping. According to the spectrum compatibility rules, TRS must envelop the RRS based on a maximum one-sixth-octave bandwidth resolution over the frequency range from 1.3 to 33.3 Hz. It is recommended that the TRS should not exceed the RRS by more than 30% over the amplified region of the RRS. In the signal reproduced by the table in the course of the testing, TRS may not fully envelop the RRS. The general requirement for a retest may be exempted if the following criteria are met. In cases where it can be shown by the use of the resonance search that no resonance response phenomena exist below 5 Hz, TRS is required to envelop the RRS only

down to 3.5 Hz (i.e., not along lower frequencies). In the case of resonance phenomena for frequencies lower than 5 Hz, TRS can envelop the RRS only down to 75% of the lowest resonance frequency. A maximum of two of the one-sixth-octave analysis TRS points may fall below RRS for each frequency range (i.e., frequencies less than or equal to 8.3 Hz and greater than 8.3 Hz) by 10% or less, if the adjacent one-sixth-octave points are not lower than RRS ordinates.

AC156 protocol may also be used as a multi-floor dynamic testing protocol as performed in [44,45,51,87] and described in Section 4.4. This extension could also be applied to other single-floor dynamic testing protocols, implementing procedures consistent with [44,45,51,87].

### 4.3.3. BS ISO 13033

The British Standards Institution [25] defines the procedures for the definition of the seismic actions and verification of NE seismic capacity within ISO 13033 standard. The ISO 13033 standard does not specifically cover industrial facilities, including nuclear power plants, since these are addressed by other International Standards. However, the principles in this standard can be applied to the definition of seismic actions for NEs in such facilities. This code establishes that evaluation of NEs for seismic actions is required when any of the following conditions applies: (a) NEs pose a falling hazard; (b) failure of NEs can impede the evacuation of the building; (c) NEs contain hazardous materials; (d) NEs are necessary to the functioning of essential facilities after the event; and e) damage to NEs represents a significant financial loss. The behavior of NEs shall be evaluated and verified against the specified performance objectives for the ultimate limit state (ULS) and the service limit state (SLS). The following verification methods are allowed: (a) design analysis; (b) seismic qualification testing; (c) procedures that determine acceptable seismic capacity based on documented experience from past earthquakes (experience data); and (d) a combination of (a), (b), and (c).

To verify the adequacy of NEs by design analysis, a structural analysis of NEs should be performed, including their anchorage and bracing, considering using the design lateral forces defined for ULS and SLS. Each member and connection force resulting from the analysis should be compared with the design capacity of NE individual member, connection brace, or anchorage, provided by regional and national regulations and codes to verify that the capacity exceeds the demand. The design lateral seismic force of NEs attached at level i of the building structure for ULS, $F_{D,p,u,i}$, is determined as reported in Equation (6):

$$F_{D,p,u,i} = \gamma_{n,E,p} \cdot k_{D,p} \cdot F_{E,p,u,i}. \tag{6}$$

The design lateral seismic force of NEs attached at level *i* of the building structure for SLS, $F_{D,p,e,i}$, is determined according to Equation (7):

$$F_{D,p,s,i} = \gamma_{n,E,p} \cdot F_{E,p,s,i}, \tag{7}$$

where $F_{E,p,i}$ is the lateral design seismic force of the NEs attached at level *i* of the building structure for ULS or SLS; $\gamma_{n,E,p}$ is the importance factor related to the required seismic reliability of the NEs; $k_{D,p}$ is NE response modification factor to be specified according to its ductility and overstrength.

The elastic equivalent static seismic forces for ULS and SLS earthquake levels are given as Equation (8) reports:

$$F_{E,p,i\ (u\ or\ s)} = k_{I(u\ or\ s)} \cdot k_{H,i} \cdot k_{R,p} \cdot F_{G,p}, \tag{8}$$

where, $k_{I(u\ or\ s)}$ is the ground motion intensity factor to be provided by regional and national standards; $k_{H,i}$ is the floor response amplification factor at the attachment at level *i*; $k_{R,p}$ is the NE amplification factor considering the effect of the natural periods of the NEs and the building; $F_{G,p}$ is the weight (*mg*) on the NEs.

The ground motion intensity factor corresponds to that used for the supporting building. The ground motion intensity factor ($k_{I(u \; or \; s)}$) is given by Equation (9):

$$k_{I(u \; or \; s)} = k_Z \cdot k_{E,(u \; or \; s)}, \tag{9}$$

where, $k_Z$ is the seismic zoning factor and $k_{E,(u \; or \; s)}$ is the seismic ground motion intensity for ULS or SLS.

The floor response amplification factor of the building structure at the attachment location at level $i$ ($k_{H,i}$) is related to the ratio between the maximum floor acceleration over the height of the building and the zero-period acceleration at the base of the building. This factor is primarily a function of: (a) the natural periods of vibration of the building structure; (b) the type of building lateral-load resisting system; (c) the relative location of the point of attachment of the NEs to the average roof elevation of the structure with respect to grade elevation; and (d) the inherent damping and degree of inelastic behavior of the building structure which is dependent on the severity of the ground motion. A trapezoidal distribution of floor accelerations within the supporting building may be assumed when simplified static analysis procedures are implemented. This trapezoidal distribution is expressed by Equation (10):

$$k_{H,i} = \left[ 1 + \alpha \frac{z_i}{H} \right], \tag{10}$$

where $\alpha$ is a parameter that is a function of the type of lateral-load resisting system ($\alpha \leq 2.5$); $i$ is the level in the building structure of the point of attachment of the NEs with respect to grade elevation ($0 \leq i/H \leq 1.0$); $z_i$ is the elevation of level $i$ with respect to grade elevation; $H$ is the average roof elevation of the structure with respect to grade elevation.

The verification of the capacity of NEs by shake table testing is accomplished by subjecting the component to either computed or simulated elastic demand floor motions that are compatible with the floor response spectra determined by RRS. The floor response spectrum is the acceleration response spectrum at the point of NEs attachment. This may be used for NEs with natural frequencies greater than a minimum value, $f_0$, e.g., reasonably assumed to be between 1.3 and 2.5 Hz. A floor response spectrum is typically obtained from dynamic analysis of the building structure. Alternatively, for a given component frequency, the floor response spectrum ordinate may be estimated as the ratio of the seismic force at level $i$ of the building structure ($F_{E,p,i}$) to the weight of the element ($F_{G,p}$), assuming the importance factor ($\gamma_{n,E,p}$) and the NE response modification factor ($k_{D,p}$) equal to one. In particular, using the previous equations, the ordinates of the normalized horizontal floor response spectrum for a NE located at level i of the building structure can be determined as given by Equation (11):

$$A_i = F_{E,p,i}/F_{G,p} = k_{I(u \; or \; s)} \cdot k_{H,i} \cdot k_{R,p}, \tag{11}$$

where $A_i \leq A_{flexible}$ for NEs with first-mode frequencies less than $f_2$ (assumed to be a value between 10 to 16.67 Hz), and $A_i \geq A_{rigid}$ for NEs with first-mode frequencies greater than $f_2$.

$A_{flexible}$ and $A_{rigid}$ are determined based on information on the building and NE dynamic characteristics. In particular, in case there is not information regarding the building dynamic properties, these parameters can be estimated according to Equations (12) and (13):

$$A_{flexible} = k_{I(u \; or \; s)} \cdot k_{H,i} \cdot k_{R,p,flexible}, \tag{12}$$

$$A_{rigid} = k_{I(u \; or \; s)} \cdot k_{H,i} \cdot k_{R,p,rigid}, \tag{13}$$

where $k_{H,i}$ is the floor response amplification factor given by Equation (9); $k_{R,p,flexible}$ is the NE amplification factor for flexible NEs ($k_{R,p,flexible} > 1.0$); and $k_{R,p,rigid}$ is the NE amplification factor for rigid NEs ($k_{R,p,rigid} = 1.0$).

When the building dynamic characteristics are known, Equations (14) and (15) can be used:

$$A_{flexible} = A_{D,I} \cdot k_{R,p,i,flexible},$$ (14)

$$A_{rigid} = A_{D,I} \cdot k_{R,p,i,rigid},$$ (15)

where, $A_{D,I}$ is the acceleration at level i estimated via dynamic analyses (including torsional response) that utilize an elastic ground motion response spectrum or time-history analysis. *A* representative damping ratio for this spectrum is 5%.

For determining the vertical response, a fraction of the values from Equation (10) may be used, with $k_{H,i}$ evaluated at grade level for all elevations, i.e., $z/h$ equal to zero. For a given NE frequency, the ratio of vertical to horizontal response can be represented by the parameter β, assumed to vary from 1/2 to 2/3. Figure 13 shows RRS related to horizontal and vertical directions according to ISO 13033. The plateau of the RRS extends up to frequency value $f_1$ (assumed to be a value between 7.5 and 8.3 Hz), whereas the ordinate of the normalized floor response spectrum is equal to $A_{rigid}$ at frequency $f_3$ (assumed to be 33 Hz).

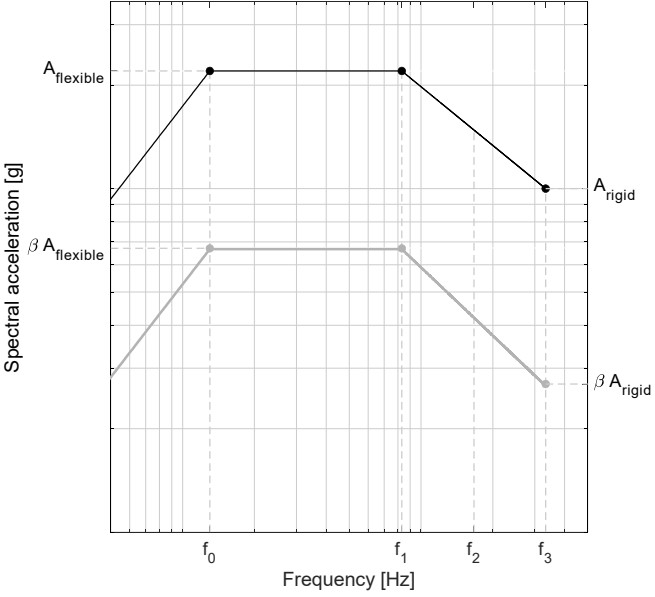

**Figure 13.** Required response spectra in the horizontal (black) and vertical (grey) directions according to ISO 13033, 5% damped.

### 4.3.4. IEEE 693

The Institute of Electrical and Electronic Engineers and Power & Energy Society (IEEE PES) developed IEEE 693 guidelines [88], which report recommendations for seismic design of substation buildings and structures and seismic design and qualification of substation equipment (i.e., NEs). This code establishes standard methods of providing and validating the seismic capacity and performance of electrical substation equipment. It provides detailed test and analysis methods for selected common equipment types or elements found in substations. The IEEE 693 guidelines are also intended to provide guidance to the manufacturers of substation equipment regarding seismic design, with regard to documentation and technical aspects associated with seismic capacity assessment and standardization purposes.

The IEEE 693 guidelines define two qualification approaches: the performance level qualification approach and the design level qualification approach. The response spectra ordinates related to the design level approach are assumed to be half of the performance level ones at any given frequency and level of damping. High, moderate, and low seismic qualification levels are defined for both approaches. Qualification levels are closely related

to ZPA, which is assumed to be the acceleration at 33 Hz or greater. For the high qualification level, horizontal ZPA associated with the seismic qualification objective is 1.0 g, and the response spectrum associated with the high-performance level is obtained by Equation (16):

$$
S_a = \begin{cases} 2.288\,\beta\,f & for\ 0.0 \le f \le 1.1\ Hz \\ 2.50\,\beta & for\ 1.1 < f \le 8.0\ Hz \\ \frac{(26.4\,\beta - 10.56)}{f} - 0.8\,\beta + 1.32 & for\ 8.0 < f \le 33\ Hz \\ 1.0 & for\ f > 33\ Hz \end{cases},
\tag{16}
$$

where $\beta$ is a function of the damping coefficient expressed as a percentage ($d \le 20\%$) and evaluated through Equation (17):

$$
\beta = \frac{3.21 - 0.68\ln(d)}{2.1156}.
\tag{17}
$$

For the moderate qualification level, ZPA associated with the seismic qualification objective is 0.5 g and the related response spectrum is assumed to be half the related high qualification level. For the low qualification level, there is no horizontal ZPA associated with the seismic qualification level. The low seismic level represents the performance level that can be expected when relatively adequate construction and seismic installation practices are used, when no special consideration is given to the seismic performance of the equipment. The selection of the seismic qualification level is a responsibility of the user and is normally based on an assessment of site geophysical parameters, risk assessments, and economics. The RRS does not include the influence of the dynamic characteristics of the building response. Therefore, one of the following alternatives may be used to account for the effects of building response: (1) defining a 2% damped response spectrum that represents the position-specific response within the building to the elastic design spectrum as determined according to the building code; (2) multiplying the RRS by a factor of 2.5. Figure 14 shows the RRS for high and moderate seismic performance levels and 5% damped.

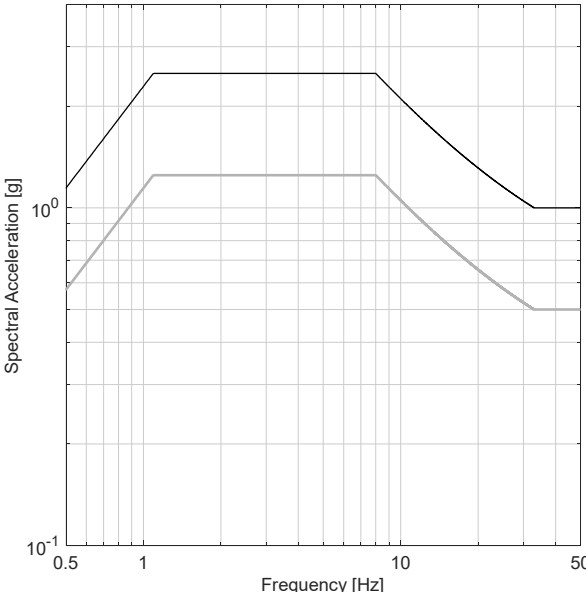

**Figure 14.** Response spectrum required according to IEEE 693 for high (black) and moderate (grey) seismic performance levels, 5% damped.

The equipment or element should be tested/analyzed in its equivalent in-service configuration, including supporting systems. When this is not possible (i.e., in situations that are not practical or economical), a modified input motion or dynamically equivalent

structure can be considered. Similarly to other codes/standards, IEEE 693 defines two types of tests: (a) resonant frequency search tests and (b) seismic simulation tests. A sine sweep or random noise excitation test shall be used for the frequency search test; frequency search above 33 Hz is not required. No resonant frequency search in the vertical axis is required if it can be shown that no resonant frequencies exist below 33 Hz in the vertical direction. Regarding seismic simulation tests, the test time histories shall be triaxial with the simulation of translational ground accelerations in three orthogonal directions. The TRS shall envelop RRS along the two perpendicular horizontal and vertical axes of the equipment, with a response spectrum in the vertical axis that shall have an acceleration of 80% of that in the horizontal axes. The input signal of the seismic simulation tests shall have a duration of at least 20 s of strong motion. Acceleration ramp-up time and decay time shall not be included in the 20 s of strong motion. The duration of strong motion shall be defined as the time interval between when the plot of the time history reaches 25% of the maximum amplitude to the time when it falls for the last time to 25% of the maximum amplitude. The theoretical TRS shall be computed at 5% damping and shall include the lower corner point frequency of the RRS (1.1 Hz), for comparison with the RRS. The spectrum matching procedure should be conducted at 24 divisions per octave resolution or higher and shall result in a theoretical response spectrum that is within ±10% of the RRS at 5% damping. The strong part ratio of the table input motion record shall be at least 30% of the total motion duration.

When required to satisfy the operating limits of the shake table, the theoretical input motion record used for testing may be high-pass filtered at frequencies less than or equal to 70% of the lowest frequency of the NE, but not higher than 2 Hz. The table output TRS shall envelop the RRS within a −10%/+50% tolerance band at 12 divisions per octave resolution or higher. A −10% deviation is allowed, provided that the width of the deviation on the frequency scale, measured at the RRS, is not more than 12% of the center frequency of the deviation, and not more than five deviations occur at the stated resolution. For equipment that responds to a single dominant frequency in a given direction, such as instrument transformers, surge arresters, and bushings, TRS spectral acceleration at the equipment as-installed frequency shall not be less than the RRS. Over-testing that exceeds the +50% limit is acceptable in agreement with the equipment manufacturer. Exceedance of the stated upper tolerance limit at frequencies above 15 Hz is generally not of interest and should be accepted unless resonant frequencies are identified in that range.

### 4.3.5. IEEE 344

The IEEE PES provides methods and documentation requirements for seismic qualification of equipment for nuclear power-generating stations. In particular, IEEE standard 344 [24] distinguishes two categories of equipment: seismic Category I and seismic Category II equipment. Seismic Category I equipment is safety-related equipment designed to withstand the effects of a safe shutdown earthquake (SSE) and maintains the specified design function and structural integrity. Seismic Category II equipment is equipment that is not required to function but whose failure could adversely affect the safe shutdown of any Seismic Category I equipment, or could result in incapacitating injury to occupants of the control room, which is designed and constructed so that SSE would not cause a failure.

A SSE is an earthquake that is based upon an evaluation of the maximum earthquake potential considering the regional and local geology and seismology and specific characteristics of local subsurface material. A SSE would produce the maximum vibratory ground motion for which certain structures, systems, and elements are designed to remain functional. These structures, systems, and elements are those necessary to provide reasonable assurance of the following: (a) integrity of the reactor coolant pressure boundary; (b) capability to shut down the reactor and maintain it in a safe shutdown condition; (c) capability to prevent or mitigate the consequences of accidents that could result in potential offsite exposure comparable to applicable regulatory requirements.

The IEEE standard 344 considers multiple methods for seismic qualification purposes, which are grouped into four general categories: (a) predict the equipment's performance by analysis; (b) test the equipment under simulated seismic conditions; (c) qualify the equipment by a combination of test and analysis; (d) qualify the equipment through the use of experience data. Each of the above-mentioned methods, or other justifiable methods, may be adequate to verify the ability of the equipment to meet the seismic qualification requirements. The choice should be based on the practicality of the method for the type, size, shape, and complexity of the equipment configuration, whether the safety function can be assessed in terms of operability or structural integrity alone, and based on the robustness of the conclusions.

This standard includes exploratory vibration tests that are generally not part of the seismic qualification requirements but may be run on equipment to aid in the determination of the best test method for qualification or to determine the dynamic characteristics of the equipment. Moreover, the test methods for seismic qualification of the equipment generally fall into three major categories: proof testing, generic testing, and fragility testing. Proof testing is used to qualify equipment for a particular requirement. The equipment shall be subjected to the particular response spectrum, time history, or other parameters defined for the mounting location of the equipment. The equipment is tested to the specified performance requirement and not to its ultimate capability. Generic testing may be considered a special case of proof testing. The objective is to show qualification for a wide variety of applications during one test. The resultant generic RRS typically encompasses a wide frequency bandwidth with relatively high acceleration levels. Finally, fragility testing is used to determine the ultimate capacities of the equipment.

The time history of the input signal for different methods should be stationary. In order to consider the vibration build-up and low-cycle fatigue effects, the duration of the strong motion portion of each test should at least be equal to the strong motion portion of the original time history used to obtain the RRS, having at least a duration of 15 s. In case artificial seismic signals are used, the stationary part of the test defines the strong motion length. The shake table maximum peak acceleration must be at least equal to ZPA of RRS. The TRS must envelop RRS within the frequency range associated with an adequately conservative test-table motion. A 5% damping value is normally assumed. The damping should be equal to or greater than one associated with RRS one, implementing an analysis based on one-sixth-octave (or narrower) points. According to IEEE344, if the input fully envelopes the RRS, the response might be associated with higher accelerations corresponding to the lowest frequencies, and this condition often requires relatively large table displacement capacities. Accordingly, the standard proposes that the general requirement for enveloping RRS by TRS can be modified as described in the following: (a) If it can be shown by a resonance search that no resonance response phenomena exist below 5 Hz, it is required to envelop the RRS only down to 3.5 Hz. However, excitation must continue to be maintained in the 1 to 3.5 Hz range, also compliance with the capability of the test facility; (b) If there are resonance phenomena for frequencies lower than 5 Hz, the RRS can be enveloped down to 70% of the lowest resonance frequency. High-pass filtering could be used to implement this action (matching the complete RRS) or the RRS ordinates could be reduced corresponding to the lower frequencies, in order to define a motion without the lower frequencies [89].

### 4.3.6. GR-63-CORE Telcordia (Ex-Bellcore)

The GR-63-CORE testing protocol [90] presents methods, criteria, and rules for seismic tests of telecommunications equipment and systems. During an earthquake, telecommunications equipment is subjected to motions that can over-stress equipment framework, circuit boards, and connectors. The seismic motion and resulting stress on NE depend on the structural characteristics of the building/facility in which NE is contained and the severity of the earthquake. The GR-63-CORE testing protocol shows the map of earthquake risk zones in the U.S. area. In particular, five earthquake risk zones are identified (from 0

to 4, corresponding to no substantial to maximum earthquake risk). The earthquake risk zones are correlated with the expected Richter Magnitude, Modified Mercalli Index, and the expected ground and building accelerations. Seismic qualification tests established by GR-63-CORE follow an approach that takes into account the earthquake risk zone in which the NEs are installed. Thus, NEs that are in earthquake risk zone 4 will need to have a higher level of seismic performance than those that are in lower earthquake risk zones.

The telecommunications equipment shall be tested using a shake table. The acceleration time histories waveform generated by this testing protocol were synthesized from several typical earthquakes and for different building and soil site conditions. The shaking shall be applied in each of the three orthogonal directions of the test specimen, to simulate the conditions that would be encountered in service when building floors apply earthquake motions to the equipment. The TRS shall meet or exceed RRS in the frequency range from 1.0 to 50 Hz, considering 2% damping. Moreover, TRS should not exceed RRS by more than 30% in the frequency range of 1 to 7 Hz. A test may be invalid if an equipment failure occurs when the TRS exceeds the RRS by more than 30% in this frequency range. The cutoff of the high-pass filter on the drive signal shall not exceed 0.20 Hz, while the cutoff of the low-pass filter on the drive signal shall not be below 50 Hz. The TRS shall be verified at one-sixth-octave (logarithmically spaced) frequencies from 0.5 to 50 Hz. If a digital analyzer is used, the digitizing rate shall be larger than or equal to 200 Hz with a total storage capacity larger than or equal to 30 s, in real-time. The GR-63-CORE defines RRS as a function of the earthquake risk zones considered for seismic qualification of NEs. Figure 15 shows RRS for the four earthquake risk zones.

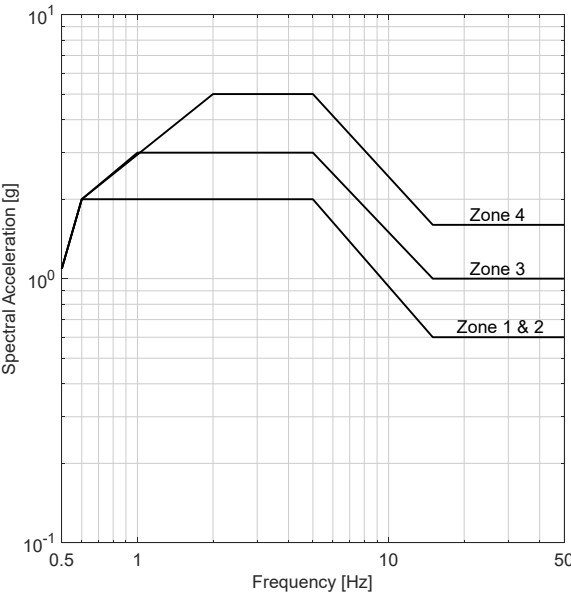

**Figure 15.** Required response spectrum according to GR-63-CORE for the four earthquake risk zones, 2% damped.

### 4.3.7. RG 1.60

Regulatory Guide 1.60 [91] of the U.S. Nuclear Regulatory Commission (USNRC) establishes a guideline to define target spectra for seismic design of structures, systems, and elements of nuclear power plants. The horizontal and vertical RRS are related to a PGA of 1.0 g and peak ground displacement (PGD) equal to 0.91 m. For different site conditions, RRS should be linearly scaled in proportion to the specified PGA or developed individually, according to the site characteristics (e.g., if the soil site has physical characteristics that could significantly affect the spectral pattern of input motion or in the occurrence of near-field ground motion). The RRS are provided for different values of the damping ratio (i.e., 0.5%, 2.0%, 5.0%, 7.0%, and 10%). A linear interpolation should be used for values in between the

provided ones. Figure 16 shows RRS related to PGA of 1.0 g and 5% damped. Regulatory Guide 1.60 does not provide criteria regarding the spectrum matching procedure and definition of input motions for testing.

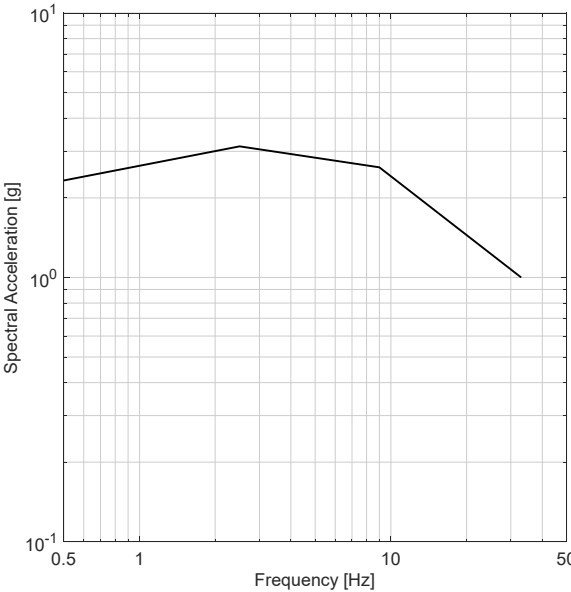

**Figure 16.** Response spectrum required according to RG 1.60 related to the PGA of 1.0 g, 5% damped.

### 4.3.8. IEC 60068

The international standard IEC 60068-2-57 [92] defines methods and criteria for testing elements, equipment, and electrotechnical products including the testing procedure for seismic applications. The standard outlines the general criteria for seismic testing described in a separate standard, IEC 60068-3-3 [93]. The procedures and methods can also be applied to other elements, and it is intended to evaluate the seismic performance of NEs during an earthquake.

The code defines two seismic classes: a general and a specific seismic class. The specific class is considered when high-reliability safety equipment for a specified environment is required (i.e., equipment in nuclear power plants); otherwise, the general class can be referred to. Equipment service conditions (e.g., electrical, mechanical, thermal pressure, etc.), and the influence of connections, cables, and piping, should be replicated in the seismic tests for both classes, or their absence justified. Moreover, qualification criteria are provided for classifying the equipment, i.e., (a) they experienced no malfunction either during or after the test, (b) they suffered a malfunction during the test but reverted to their correct state after the test, or (c) they experienced a malfunction during the test and required resetting or adjustment on completion of the test but required no replacement or repair.

The IEC 60068-2-57 includes different seismic inputs for seismic testing. In particular, the test inputs are divided into two categories: multifrequency and single-frequency waves. The test waves should (a) produce a TRS larger than or equal to RRS, (b) possess a maximum peak acceleration value at least equal to the ZPA value, (c) reproduce, with a safety margin, the effects of the required earthquake, and (d) ideally not include any frequency greater than 35 Hz. The time history obtained according to IEC 60068 shall be generated by the composition of frequencies included within a frequency range from 1 to 35 Hz. In some cases, the test frequency range may be extended or reduced depending on the effective value of the cut-off frequency of the ground response spectrum or the critical frequencies of the specimen, but this should be justified. The total duration of the time history shall be about 30 s, of which the strong part shall not be less than 20 s. Three RRS are defined, associated with 2%, 5%, and 10% damping ratio. These RRS have a generalized form that is

based on simple correlations among the corner frequencies, depending on assumptions regarding the frequency range of sensitivity of NEs.

The test should be performed through triaxial tests with input motions applied simultaneously along all principal axes of the test specimen, but this does not exclude single-axis or biaxial testing. RRS ordinates associated with the vertical direction of excitation should be 50% of the horizontal RRS ones. The IEC 60068 establishes that spectrum compatibility shall be checked in the specified range at least in one-sixth-octave bandwidth resolution in the general case, i.e., specimen damping lying between 2% and 10%. The tolerance to be applied to RRS shall be in a range between 0% and 50%. Moreover, for frequencies larger than the plateau zone, a tolerance of more than 50% is permitted. Figure 17 shows RRS related to a frequency range from 1 ($f_1$) to 35 Hz ($f_2$) and 5% damped.

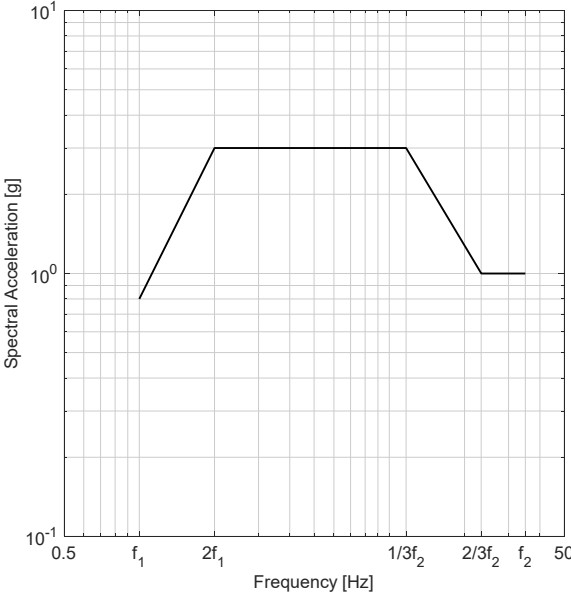

**Figure 17.** Response spectrum required according to IEC 60068 considering a frequency range from 1 ($f_1$) to 35 Hz ($f_2$), 5% damped.

### 4.4. Multi-Floor Dynamic Testing Protocols

Until recently, testing facilities could not easily implement accurate full-scale testing on NEs by means of simultaneous floor acceleration and story deformation loading conditions, reproducing the actual arrangement of NEs housed within multistory buildings and sensitive to both accelerations and displacements (e.g., partition walls, cladding curtain walls, distributed duct, piping, electrical systems, Heating, Ventilation and Air Conditioning (HVAC) systems, suspended ceilings, and ceiling-mounted equipment).

A brilliant solution, as already discussed in this article, was found in [44,45,51,87], where the multi-floor dynamic testing was carried out by means of a test frame fixed to the shake table that simulated the seismic behavior of a generic story of a building, in which the NEs (i.e., partitions) were installed. The testing setup implemented by Magliulo et al. [44] is depicted in Figure 18. The test frame was designed in order to have an assigned drift, i.e., an assigned peak displacement of the top floor, and given a peak acceleration at the bottom floor, i.e., at the shake table. The geometry of the test frame was defined taking into account three requirements: (i) realistic value of mass; (ii) realistic interstory height h, assumed equal to 2.74 m; (iii) realistic interstory displacement $d_r$, assumed equal to 0.005 h for a Damage Limit State (DLS) earthquake with 50-year return period. This strategy allows the assigning at the bottom floor of a time history according to single-floor dynamic testing protocols, e.g., AC156.

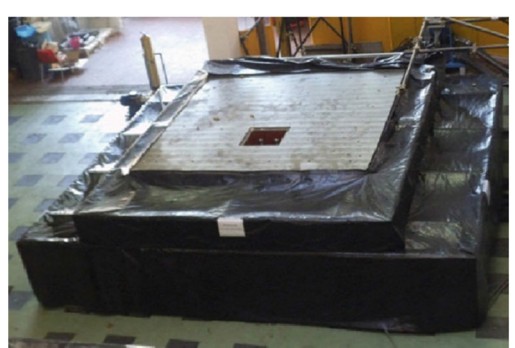 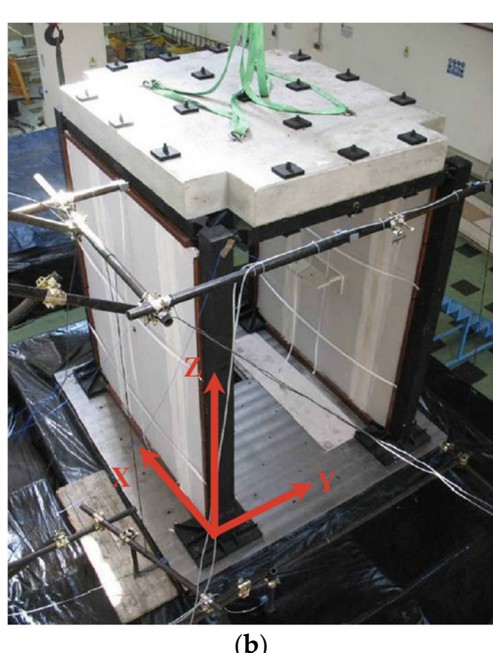

(**a**) (**b**)

**Figure 18.** Test setup for multi-floor dynamic testing according to Magliulo et al.: (**a**) shake table test and (**b**) test frame [44].

A different solution was found by the University at Buffalo, which commissioned a dedicated Nonstructural Component Simulator (UB-NCS) composed of a two-level testing frame capable of simultaneously subjecting NEs sensitive to both accelerations and displacements to realistic full-scale floor motions expected within multistory buildings also applying story deformation loading conditions. The relevant international reference testing and qualification protocol was developed by Retamales et al. [94]; a representative testing setup is shown in Figure 19 [94]. The protocol reduces the minimum testing frequency that can be considered in an experiment from 1.3 Hz, as in the current AC156 procedure, to 0.2 Hz, allowing for more realistic testing of NEs sensitive to low-frequency excitations (e.g., tall slender cantilever-type equipment, base-isolated equipment, etc.). The objective of this protocol is not to replace current testing protocols, but rather to enhance the capabilities and the type of equipment that can be tested. Unlike other testing protocols, this protocol is presented as a set of closed-form expressions defining a pair of displacement histories for the bottom and top levels of the UB-NCS that simultaneously matches: (i) a target acceleration response spectrum (either ground or floor response spectra), and (ii) a target-generalized interstory drift. Both the target spectral accelerations and drifts can be specified based on the expected values at a given normalized building height h/H, where h is the NE installation height above grade and H is the total building height. This qualification testing protocol considers as variables: (i) the location of the NEs along the height of the building through the parameter h/H; (ii) the range of frequencies to be assessed during testing ($f_{min}$–$f_{max}$); and (iii) the ASCE 7-16 design spectral response in the short period range, $S_{DS}$, and corresponding to 1 s period, $S_{D1}$. The frequency content targeted for the seismic qualification testing protocol covers the range of frequencies between $f_{min}$ = 0.2 Hz and $f_{max}$ = 5 Hz. This range corresponds to the operating frequencies of the UB-NCS and the expected fundamental periods of typical multistory buildings, including some higher vibration modes.

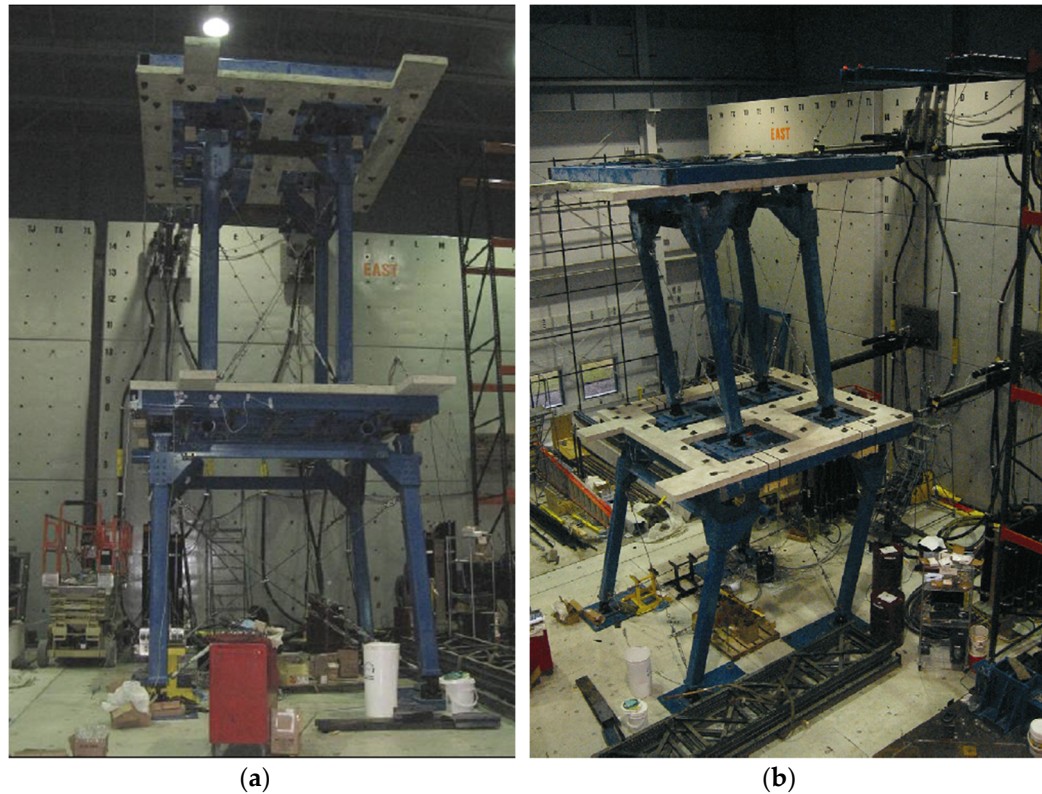

**Figure 19.** Test setup for multi-floor dynamic testing according to Retamales et al.: (**a**) front view and (**b**) isometric view [94].

## 5. Novel Perspectives toward a Unified Testing Approach

### 5.1. Criticalities of Existing Protocols

The test protocols reviewed here are widely used in the literature, especially AC156 and FEMA 461. The AC156, FEMA 461 and ISO 13033 protocols were developed to assess and/or qualify the seismic performance of generic elements, and, among them, the AC156 protocol is the only one explicitly aimed at seismic certification. However, few recent research studies have pointed out potential criticalities of the AC156 protocol, in terms of both spectral demands (RRS) [66,95–97] and damage severity (e.g., for peculiar applications) [98]. The other protocols are intended for systems that are housed within critical facilities (i.e., telecommunications, electrical and nuclear systems) and provide a more standardized qualification approach, e.g., regarding the target performance levels or the seismic zones. Most protocols do not clearly define the applicability conditions with regard to the damage and response sensitivity of NEs. Overall, the aim of the reference protocols is often to assess whether the tested element, subject to a target seismic event (i.e., artificial or natural earthquake), meets certain functionality or stability requirements, i.e., pass or fail qualification outcomes. This criterion can be appropriate when the site and the element/building properties are known, which is peculiar to specific NEs (e.g., critical electric equipment for power stations). In different conditions, such as the generic qualification of NEs by the manufacturers, this approach cannot be easily applied, and the seismic evaluation of the element cannot be generalized.

### 5.2. Potential Improvement Interventions

Current regulations and codes often require the seismic design and safety verification of NEs. For this purpose, test protocols should include criteria and rules for the estimation of the dynamic properties (DPs) and the significant performance parameters (s) of the NEs to be tested. Dynamic properties are mostly required for a relatively accurate evaluation of the seismic demand, in compliance with the regulations/codes (e.g., [11,99]); DPs generally include the fundamental period ($T_a$), the damping ratio ($\xi_a$), and the (acceleration) amplification factor ($a_p$) of the NE.

Significant performance parameters represent the quantitative parameters, or measures, to be considered for assessing seismic demands on and capacity of NEs and to verify the safety conditions. The SPPs can be considered to be efficient and applicable EDPs, with regard to the specific applications and testing procedures. As was discussed in the previous sections, these parameters depend on (a) the response and damage sensitivity of NEs to seismic actions, including site, building, and relevant interaction responses, and (b) the possibility of robustly assessing them through consolidated experimental testing procedures. The SPPs can be selected among inertial or deformation measures, or by the combination of them, according to the response/damage mechanisms of interest of the NEs to be tested. Accordingly, both seismic capacity and demand should be estimated considering consistent SPPs. The experimental testing procedures reviewed in this study can be used to assess the seismic capacities. The relevant codes/regulations typically define approaches and formulations for defining the seismic demand, which may also depend on the dynamic properties of both buildings and NEs, as was discussed in the previous sections.

For acceleration-sensitive NEs, the seismic capacity can be expressed in terms of peak floor acceleration (PFA) or peak component acceleration (PCA). The PFA is the maximum acceleration obtained on the floor on which the element was installed, while PCA is the maximum acceleration recorded on the component (e.g., in the element's center of mass during the tests). For displacement- or deformation-sensitive elements, the seismic capacity can be expressed in terms of interstory drift ratio (IDR) or target displacement or deformation measures ($\delta$), depending on the applied NE deformations that are relevant to the damage. Finally, for NEs that are sensitive to both acceleration- and deformation-based measures, both types of SPPs should be considered, also accounting for the relevant damage response/mechanisms.

### 5.3. Technical Recommendations and Final Remarks

Technical recommendations are developed for implementing a unified approach for seismic assessment of NEs by means of experimental tests, including seismic qualification purposes. These recommendations were defined in light of the critical review and assessment of current methods and protocols, according to the evidence pointed out in the previous sections. The technical recommendations and innovative perspectives derived in this study aim at improving the seismic assessment and qualification of NEs by means of experimental testing and represent an original literature and practice contribution.

Table 1 summarizes these recommendations. In particular, the key parameters/features associated with the seismic assessment, qualification, and safety verification of NEs are specified for the most common and representative NEs. The key parameters/features consist of response/damage sensitivity, SPPs, DPs, and recommended test types. No other studies have provided a critical review assessment and technical-scientific guidance regarding these applicative aspects, which are essential for implementing relatively reliable and robust assessment and qualification procedures. Table 1 represents a reference for both researchers and practitioners, and it might be implemented by codes and regulations.

**Table 1.** Recommended test protocol and DPs and SPPs required for the tested element.

| Nonstructural Element | Response/Damage Sensitivity | | | SPPs | DPs | Test |
|---|---|---|---|---|---|---|
| | Acc. | Disp. | Both | | | |
| Infill walls, partitions, openings (doors, windows) Facades, glazing systems, and curtains Ceiling systems Systems inside the building (pipes carrying pressurized fluids, fire hydrant piping systems, and other fluid pipe systems) | | | √ | IDR or $\delta$, PFA, PCA | $T_a$, $\xi_a$, $a_p$ | Multi-floor dynamic or (secondarily) Quasi-static and Single-floor dynamic |
| Cabinets, storage racks, bookcases, and shelves Appliances (refrigerators, washing machines, gas cylinders, TVs, diesel generators, water pumps (small), window ACs, wall-mounted ACs) Vertical projections (chimneys and stacks, parapets, water tanks (small), hoardings anchored on rooftops, antenna communication towers on rooftops) Horizontal projections (sunshades, canopies, and marquees) Storage vessels and water heaters (flat-bottom containers and vessels, structurally supported vessels) Mechanical equipment (boilers and furnaces, general manufacturing and process machinery, HVAC equipment) Hospital cabinets, museum artifacts, and freestanding objects | √ | | | PFA, PCA | $T_a$, $\xi_a$, $a_p$ | Single-floor dynamic |
| Systems from within and from outside to inside the building (water supply pipelines, electricity cables and wires, gas pipelines, sewage pipelines, telecommunication wires, rainwater drainpipes, elevators, fire hydrant systems, air-conditioning ducts) | | √ | | IDR or $\delta$ | - | Quasi-static |

When multi-floor dynamic tests cannot be performed, quasi-static and single-floor dynamic tests can be conducted separately. An example is the testing procedure adopted by Coppola et al. [100] by means of a special testing facility at the Components and Building Systems Laboratory of the Construction Technologies Institute of the National Research Council of Italy (ITC-CNR) in San Giuliano Milanese (Italy). Specifically, the authors conducted quasi-static and dynamic tests for the seismic evaluation of an innovative cladding system. The facility is able to accommodate full-size plane elements (partition systems, infill systems, façade systems, etc.), up to 6.3 m wide and up to 8.0 m high. The components can be anchored to the steel supporting frame by means of three beams: one fixed beam at the bottom and two moving beams at the second and third levels. The intermediate and superior beams can be moved, in the plane and out of the plane direction, through six hydraulic actuators, to simulate seismic actions. A mechanical lift system for the moving beams allows for various interstory heights. The moving beams are supported on low-friction rollers and connected to a dynamically controlled hydraulic actuator system. The load cell and transducer of the hydraulic actuator relate to an advanced digital controller that enables the acquisition of real-time load and displacement data.

The testing procedure adopted by Coppola et al. [100], consists of cyclic quasi-static tests, performed along the in-plane direction according to the loading procedure proposed by FEMA 461, and incremental dynamic tests performed according to AC156.

A unified approach should not be limited to the robust selection of the key parameters and testing approaches/protocols, which already signifies a novel and crucial step in the field. In fact, the selected testing protocols should be applied by maximizing the testing outcomes in terms of systematicity and comprehensiveness. In other words, once the protocol is defined, the testing procedure and program should be implemented in order to identify and characterize the seismic response and damage of the tested NEs in a systematic and comprehensive manner. For example, (a) the dynamic properties of the specimen should be estimated corresponding to all relevant DSs, providing useful information regarding the influence of DSs on the dynamic properties, and (b) all relevant DSs, from operativity to ultimate/failure conditions, should be associated with thresholds of SPPs (NE capacity thresholds). However, these experimental correlations should be

established by means of standardized approaches, which would allow fully consistent comparisons/extensions and, potentially, generalization. Even though few protocols include directions for implementing a more general and systematic approach, no systematic and standardized recommendations are generally provided. Moreover, most protocols still recommend a pass or fail assessment/qualification approach, which represents a limited application of the potentiality of the above-mentioned approach. Therefore, further studies should address the above-mentioned issue, by providing a unified approach in terms of testing procedure application and the systematicity and comprehensiveness of the experimental assessment and qualification of NEs.

## 6. Final Remarks and Conclusions

The paper addresses an extremely critical research gap: methods and protocols for the experimental seismic assessment and qualification of nonstructural elements (NEs), considering elements that are sensitive to accelerations, displacements, and both parameters. In fact, despite the copious literature addressing the seismic testing of NEs, no literature studies have summarized and analyzed the testing and assessment methodology, and the literature studies represent peculiar case studies or fragmented applications. Furthermore, for complex but relatively common cases, e.g., NEs sensitive to both accelerations and displacements, no clear testing requirements or assessment criteria are provided in the literature, unless a few peculiar applications are considered.

In order to cover the above-mentioned literature gap, the study reviews and analyzes the latest literature studies and regulations/codes regarding seismic damage and classification of NEs, providing novel evaluation remarks. The core of the paper consists of a critical and systematic assessment of the reference international testing protocols and guidelines for seismic assessment and qualification purposes. The scope of the investigation is wide and tends to be comprehensive: quasi-static, single-floor dynamic (shake table), and multi-floor dynamic testing procedures are considered, including multiple protocols, when available. Innovative and relatively simplified solutions for testing NEs that are sensitive to both accelerations and displacement are reported and discussed. The work by Pali et al. [37] is reported as a reference application for quasi-static testing, describing the following protocols: deformation- and force-controlled FEMA 461 [82], CUREE-Caltech, AAMA 501.4 and 501.6 [84]. The shake table tests by Truong et al. [85] are described, and several testing protocols are reported: FEMA 461 [82], AC156 [26], ISO 13033 [25], IEEE 693 [88], IEEE 344 [24], GR-63-CORE [90], and RG 1.60 [91], IEC 60068 [92,93]. Two reference applications are described with regard to multi-floor tests: Magliulo et al. [44], who carried out multi-floor tests through shake table tests, and Retamales et al. [94], who developed a multi-floor testing protocol by using the UB-NCS facility of the University at Buffalo.

In light of the literature assessment, the criticalities of the existing testing protocols are highlighted, and novel perspectives are developed toward a unified testing approach. In particular, technical recommendations provide guidance for implementing reliable and robust testing procedures. In particular, the relevant parameters and features that are essential for carrying out experimental assessment and qualification procedures are defined for a wide range of NEs, also providing general rules for identifying the relevant NEs in terms of response/damage sensitivity. Furthermore, the appropriate testing method is also recommended. If NEs are acceleration-sensitive, single-floor dynamic tests should be performed, according to the relevant testing protocols, and PFA (peak floor acceleration) and PCA (peak component acceleration) ( $T_a$ (NE fundamental period), $\xi_a$ (NE damping ratio), and $a_p$ (acceleration amplification factor)) can be considered as significant performance parameters (SPPs) (dynamic parameters (DPs)). In the case of displacement-sensitive elements, quasi-static tests should be carried out, following the relevant protocols, and IDR (interstory drift ratio) or $\delta$ (deformation parameter) can be considered as s. The NEs that are sensitive to both accelerations and displacements/deformations should be assessed via multi-floor dynamic tests or, when not possible, through both quasi-static testing and

single-floor dynamic tests; reference studies are provided to guide these tests [44,49,100]; in this case, PFA, PCA, IDR or δ ( $T_a$, $ξ_a$, and $a_p$) can be considered as s (DPs). The technical information and evaluation remarks provided regarding the available protocols might be useful for selecting the most appropriate testing protocols. In practical terms, the most relevant and/or common NEs have been associated with specific testing methods, also referring to optimal testing protocols. Alternative testing approaches, including innovative testing setups (and related scientific references), have been reported and discussed, representing operative guidelines, possibly to be implemented by regulations and codes.

To conclude, the paper contributes to the literature in terms of two key outcomes: technical-scientific review, and technical recommendations (unified testing approach). To the knowledge of the authors, no other studies have carried out a review assessment of the reference testing methods and protocols. Conversely, this study presents a systematic and comprehensive review that allows the identification of the criticalities and strengths of the available codes/protocols. The technical recommendations provided in the paper lay the groundwork for a more robust and standardized testing and qualification framework. In particular, the provided data might represent the first step in developing code and regulation criteria for the seismic assessment and qualification of NEs by means of experimental methods.

**Author Contributions:** Conceptualization, M.Z., R.N., P.D., D.D. and G.M.; Data curation, M.Z. and D.D.; Formal analysis, R.N., D.D. and G.M.; Funding acquisition, R.N., P.D., D.D. and G.M.; Methodology, M.Z., R.N., P.D., D.D. and G.M.; Project administration, P.D.; Resources, R.N., P.D., D.D. and G.M.; Software, M.Z. and D.D.; Supervision, R.N. and G.M.; Validation, R.N. and G.M.; Writing—original draft, M.Z.; Writing—review and editing, R.N., P.D., D.D. and G.M. All authors have read and agreed to the published version of the manuscript.

**Funding:** This research was funded by the Industrial research project CADS "Creazione di un Ambiente Domestico Sicuro" and by the Italian Department of Civil Protection (DCP) in the frame of the national project DPC–ReLUIS 2022-2024 WP 17: "Code contributions for nonstructural elements". The article processing charge (APC) costs were covered by the Travel Award sponsored by the open access journal Buildings published by MDPI.

**Institutional Review Board Statement:** Not applicable.

**Informed Consent Statement:** Not applicable.

**Data Availability Statement:** The data presented in this study are available on request from the corresponding author.

**Conflicts of Interest:** The authors declare no conflict of interest. The funders had no role in the design of the study; in the collection, analyses, or interpretation of data; in the writing of the manuscript; or in the decision to publish the results.

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
