# Peer review of "Experimental Seismic Assessment of Nonstructural Elements: Testing Protocols and Novel Perspectives"

_buildings, doi:10.3390/buildings12111871_

Round 1

Reviewer 1 Report

Article represents a comprehensive review. However, the proposed test protocols in Table 1 should be more specific, not just general. Per example limit values for separate nonstructural element and coresponding DPs could be added. I also would recommend that native English speaker review's the text.

Author Response

Please, find attached the responses to Referee 1.

Reviewer 2 Report

The article raises a very important topic of non-structural elements and their role in the dynamical response of the structure. The problem is often neglected by designers although it could decide of life and death of the buildings residents which even turned out after the earthquake in Turkey in 1999. Therefore it is very good that the authors have dealt with this topic. But unfortunately the article is not as interesting as it could be. First of all the tittle suggests that  we get an experiment with its assumptions, course of action, many experimental photographs, a lot of in-situ results. Unfortunately we get none of this. In return, we get a review, which, unfortunately, is poorly described, mixed with something like a simulation.

I will try to provide detailed comments in points:

·         The illustrations, tables are clear.  

·         Many references are grouped together ex. [54–60],[ 65–70]. It will be better to write some sentence of each of these articles. In general, for an experimental article, these references are too many, and as for the review, they are not sufficiently described

·         I do not know why the excitations given in Fig. 3 or Fig. 4 end in the max value

·         Inputs in Fig. 8 start from high amplitude instead

·         Good response spectra procedure

·         Conclusions are not support by the results, there is no discussion in this part.

·         No highlighting of the novelty

From all these comments it follows that I cannot accept this article in its present form. Authors should decide whether they are doing a decent review or whether it is an article with experimental features.

Author Response

Please, find attached the responses to Reviewer 2.

Reviewer 3 Report

In this manuscript, the authors present an experimental study on nonstructural elements under earthquake. This work is interesting and worth publishing. The paper is well organized. I would like to suggest some changes before the publication of the paper. First, some photos of the experimental text device are recommended to be added in the manuscript. Second, a flowchart of this study should be added in the section Introduction.

Overall, the manuscript addresses an interesting problem and should deserve attention; however, further review is expected.

Author Response

Please, find attached the responses to Reviewer 3.

Round 2

Reviewer 2 Report

Now, the paper looks much better. The only thing he can still complain about is the lack of any reference to contemporary literature in the conclusion. The conclusion has been widened but they are still stewards with no reference to a specific source

Author Response

Dear Referee,

please, find attached the responses related to the re-revision.
